# Versatile Application of Nanocellulose: From Industry to Skin Tissue Engineering and Wound Healing

**DOI:** 10.3390/nano9020164

**Published:** 2019-01-29

**Authors:** Lucie Bacakova, Julia Pajorova, Marketa Bacakova, Anne Skogberg, Pasi Kallio, Katerina Kolarova, Vaclav Svorcik

**Affiliations:** 1Department of Biomaterials and Tissue Engineering, Institute of Physiology of the Czech Academy of Sciences, Videnska 1083, 142 20 Prague 4-Krc, Czech Republic; Julia.Pajorova@fgu.cas.cz (J.P.); Marketa.Bacakova@fgu.cas.cz (M.B.); 2BioMediTech Institute and Faculty of Medicine and Health Technology, Tampere University, Korkeakoulunkatu 3, 33720 Tampere, Finland; anne.skogberg@tuni.fi (A.S.); pasi.kallio@tuni.fi (P.K.); 3Department of Solid State Engineering, University of Chemistry and Technology Prague, Technicka 5, 166 28 Prague 6-Dejvice, Czech Republic; Katerina.Kolarova@vscht.cz (K.K.); Vaclav.Svorcik@vscht.cz (V.S.)

**Keywords:** bacterial nanocellulose, nanofibrillated cellulose, animal nanocellulose, algal nanocellulose, tissue engineering, tissue repair, wound dressing, cell delivery, drug delivery, antimicrobial properties

## Abstract

Nanocellulose is cellulose in the form of nanostructures, i.e., features not exceeding 100 nm at least in one dimension. These nanostructures include nanofibrils, found in bacterial cellulose; nanofibers, present particularly in electrospun matrices; and nanowhiskers, nanocrystals, nanorods, and nanoballs. These structures can be further assembled into bigger two-dimensional (2D) and three-dimensional (3D) nano-, micro-, and macro-structures, such as nanoplatelets, membranes, films, microparticles, and porous macroscopic matrices. There are four main sources of nanocellulose: bacteria (*Gluconacetobacter*), plants (trees, shrubs, herbs), algae (*Cladophora*), and animals (*Tunicata*). Nanocellulose has emerged for a wide range of industrial, technology, and biomedical applications, namely for adsorption, ultrafiltration, packaging, conservation of historical artifacts, thermal insulation and fire retardation, energy extraction and storage, acoustics, sensorics, controlled drug delivery, and particularly for tissue engineering. Nanocellulose is promising for use in scaffolds for engineering of blood vessels, neural tissue, bone, cartilage, liver, adipose tissue, urethra and *dura mater*, for repairing connective tissue and congenital heart defects, and for constructing contact lenses and protective barriers. This review is focused on applications of nanocellulose in skin tissue engineering and wound healing as a scaffold for cell growth, for delivering cells into wounds, and as a material for advanced wound dressings coupled with drug delivery, transparency and sensorics. Potential cytotoxicity and immunogenicity of nanocellulose are also discussed.

## 1. Introduction

Cellulose is a linear polymer of glucose and is the most abundant biopolymer on Earth. Nanocellulose can be defined as cellulose in the form of nanostructures, which are features not exceeding 100 nm at least in one dimension. In other dimensions, these structures can reach hundreds of nm, micrometers, or even more, particularly in the case of electrospun nanofibers. According to their morphology, the cellulose nanostructures can be divided into nanofibrils, nanofibers, nanowhiskers, nanocrystals, nanorods, and nanoballs (Table 1). Nanofibrils are typically present in bacterial cellulose, where they form a hydrogel [1,2,3], or they can be obtained from plants, particularly from wood, by hydrolysis, oxidation, and mechanical disintegration [4,5,6,7,8,9,10]. The term “nanofibers” is usually used for fibrous structures thicker and longer than nanofibrils, particularly structures created by electrospinning of cellulose without additives or in composites with other natural and synthetic polymers. Electrospun nanofibers are often more than 100 nm in diameter (i.e., several hundreds of nm). In fact, they are submicron-scale fibers, but the term “nanofibers” has become widely used for them (for a review, see [11]). The distinction between the terms “nanofibrils” and “nanofibers” is often unspecified. For example, some authors have referred to the nanofibrils present in bacterial cellulose as “nanofibers” [12,13,14]. Similarly, very thin fibrous cellulosic structures with characteristics of nanofibrils, isolated from pineapple, have been referred to as “nanofibers” [15]. Cellulose nanowhiskers, nanocrystals, and nanorods are also fibrous structures similar in diameter to nanofibrils, but usually shorter. Nanocrystals have a needle-like or rod-like morphology [16,17,18]; nanorods are in fact nanocrystals with a rod-like morphology [19]. In the scientific literature, the terms “cellulose nanowhiskers,” “cellulose nanocrystals,” and “cellulose nanorods” are often used as synonyms [10,20,21,22]. All these nanostructures are typically prepared by acid hydrolysis of cellulose, which destroys the amorphous regions, while the crystalline segments of the cellulose remain intact. As a result, the cellulose nanocrystals, nanowhiskers, and nanorods have a higher crystallinity index than cellulose nanofibrils, which are more typically prepared by enzymatic hydrolysis, e.g., by xylanases and cellulases [9], and by mechanical disintegration of cellulose, e.g., by grinding, homogenization and shearing [8,10]. Nanocrystalline cellulose particles can also have a spherical morphology, being referred to as nanocellulose balls or cellulose nanoballs [23,24]. Nanoplatelets are assemblies of nanofibrils into plate-like structures of nanoscale thickness but with other dimensions in micrometers [25].

Cellulose nanostructures, especially nanofibrils, can be further assembled into bigger two-dimensional (2D) and three-dimensional (3D) micro- and macro-structures. Two-dimensional structures include membranes and films in the self-supporting form [5,33] or in the form of material coatings [34,35], while 3D structures include microparticles, such as microneedles [36] and porous microbeads [37,38], and macroscopic matrices, such as porous aerogels and hydrogels, foams, and sponges [39,40,41,42].

As a natural polymer, cellulose, including nanocellulose, is usually obtained from natural sources, although industrial residues, e.g., from beer production [43], or from municipal solid wastes (*Panax ginseng*, spent tea residue, waste cotton cloth, old cardboard) are considered as new important precursors of "green" nanocellulose [44]. There are four natural sources of nanocellulose: bacteria, plants, algae, and animals. Bacterial cellulose, also known as microbial cellulose [45,46], is produced extracellularly by gram-negative bacteria of various genera, e.g., *Acetobacter, Achromobacter, Aerobacter, Agrobacterium, Alkaligenes, Azotobacter*, *Pseudomonas, Rhizobium*, *Rhodobacter, Salmonella, Sarcina,* and particularly *Gluconacetobacter*, which is the most efficient producer (for a review, see [47,48]). The most widely used species of *Gluconacetobacter* is *Gluconacetobacter xylinus* (synonyms *Acetobacter xylinus*, *Komagataeibacter xylinus*) [1,49]. Other important species include *Gluconacetobacter hansenii* [3,50,51], *Gluconacetobacter kombuchae* [52], *Komagataeibacter (Gluconacetobacter) europaeus* [53], and low pH-resistant strain *Komagataeibacter (Gluconacetobacter) medellinensis* [42]. The bacterial growth and production of nanocellulose can be further enhanced by the presence of yeasts or yeast extract in the culture medium [52,54], or by symbiotic co-cultivation with *Мedusomyces gisevii* [55]. 

Bacterial cellulose is chemically identical with plant cellulose but is free of byproducts like lignin, pectin, and hemicelluloses, featuring a unique reticulate network of fine fibers [56].

Plant nanocellulose can be obtained from abundant sources derived from trees, shrubs, various herbs, grasses, flowers, root vegetables, succulents, etc. The trees include leaved trees, e.g., birch [33,57,58,59,60,61], and various coniferous trees [26,27,62,63,64], e.g., *Pinus radiata* [65]. Other trees are *Acacia mangium* [66], balsa [67], *Syzygium cumini* [68], banana pseudostem [5], palm [7,8,69], *Khaya senegalensis* [70], and citrus trees [71]. Nanocellulose from leaved trees is usually referred to as hardwood-derived, while nanocellulose from coniferous trees is softwood-derived. Shrub sources of nanocellulose are cotton [32] and hibiscus [30,72]. Other important plant sources include sugar cane [73,74], grass, e.g., *Miscanthus Giganteus* [75] or *Imperata brasiliensis* [76], bamboo [77], rice husk [78], corn leaf [34], triticale straw [79], pineapple leaf [15], soybean straw [9], carrot [80], and agave [25], particularly *Agave sisalana*, i.e., sisal [81,82].

Algae as sources of nanocellulose are *Cladophora* [37,38,83,84,85,86,87] and *Cystoseria myrica* [88]. Nanocellulose materials derived from *Cladophora* have been tested mainly for their potential biomedical applications in terms of the presence of impurities, such as heavy metals, glucans, and endotoxins [85]. Their suitability as scaffolds for cell cultivation [84], their hemocompatibility [37], and their adsorption capacity for Congo Red dye [38] have also been evaluated. Nanocellulose derived from *Cystoseria myrica* combined with Fe_3_O_4_ has been tested for removal of mercury ion pollution [88].

Animal sources of nanocellulose include tunicates, i.e., animals belonging to the phylum *Chordata*, such as *Styela clava* [89,90,91] (for a review, see [92]) and *Halocynthia roretzi Drasche* [93]. Cellulose films derived from *Styela clava* tunics have been tested for wound dressings [90,91], and they also have potential for other biomedical applications, such as stitching fibers, scaffolds for tissue engineering, absorbable hemostats and hemodialysis membranes [89]. Animal-derived nanocellulose also has potential applications in industry and in technology. A composite nanocellulose membrane derived from *Halocynthia roretzi Drasche*, endowed with TiO_2_ nanoparticles, has been used for removing oils from wastewater [93].

Nanocellulose possesses a wide spectrum of advantageous physical, chemical, and biological properties. Its large specific surface area enables the adsorption of various atoms, ions, molecules and microbial cells, and porous nanocellulose materials are able to separate various molecules and to retain microbial objects. Nanocellulose-based materials in general have high mechanical strength, chemical inertness, and tailorable morphological, physical, chemical, electrical, thermal, and optical properties, barrier properties, and antimicrobial effects and biocompatibility with no toxicity or low toxicity and with low immunogenicity. At the same time, they are relatively low-cost materials with high availability and renewability. Nanocellulose materials have therefore emerged as promising materials for a wide range of industrial, technological, and biomedical applications, namely purification of air and aqueous solutions, filtration and ultrafiltration, packaging of food and other sensitive products, conservation of historical artifacts, construction of thermal insulators and fire retardants, energy extraction and storage, acoustics, sensorics, and controlled drug delivery. All these applications are summarized with some examples in Table 2.

Other important applications of nanocellulose, on which the following part of this review is focused more deeply, are applications in tissue engineering, tissue repair and wound healing. The main examples of these applications include engineering of blood vessels, neural tissue, bone, cartilage, liver and adipose tissue, reconstruction of urethra and *dura mater*, repairing connective tissue and congenital heart defects, constructing protective barriers and ophthalmologic applications, mainly construction of contact lenses. Other interesting applications of nanocellulose are enhancement of the efficiency of cell transfection and creation of 3D culture environment for maintaining the pluripotency of stem cells. All these applications are discussed in more details in Section 2 (“History of nanocellulose research with focus on biomedical applications”) and Section 3 (“Recent use of nanocellulose in tissue engineering and tissue repair”).

A considerable part of this review is dedicated to the application of nanocellulose in skin tissue engineering and wound healing (Section 4 and Section 5). To the best of our knowledge, there are no review specialized or at least deeply focused on the use of nanocellulose in these applications. The skin is the largest organ of the human body with several vitally important functions, particularly as barrier against adverse effects of the surrounding environment on the organism (chemical damage, radiation damage, e.g., by ultraviolet light, and microbial infection). Other important functions of skin include thermoregulation, sensation of temperature, touch, pressure and pain, keeping appropriate moisture in the underlying tissues, excretion of ions, water, and various molecules (e.g., lipids and proteins), and also production and storage of various biomolecules, such as pigments, vitamin D, and keratins for formation of epidermal appendages (for a review, see [26,138,139]). Therefore, there is essential need to regenerate or at least to repair the damaged skin, particularly by methods of skin tissue engineering and induction of active wound healing. Nanocellulose has several advantageous properties for these applications, such as appropriate mechanical strength, high water-absorbing capacity, which enables to keep the moisture in the damaged skin, and, at the same time, to absorb the exudate from the wounds [26,27,60,140,141,142,143]. The nanoscale morphology of the nanocellulose mimics the nanoscale architecture of the native extracellular matrix, and thus the nanocellulose can be regarded as a suitable substrate for the adhesion and growth of skin cells, although the non-degradability of this material is an important factor limiting its direct use as scaffolds for cells in skin tissue engineering. Last but not least, some types of nanocellulose, e.g., wood-derived nanocellulose, have antimicrobial effect [27,65], or this effect can be induced by incorporation of nanocellulose materials with various ions and compounds (for a review, see [142,144]).

Physicochemical properties of nanocellulose, such as its wettability or electrical charge, which are important for tissue engineering and other biotechnological applications of this material, can be tailored by functionalization of nanocellulose with various chemical groups or by preparation of nanocellulose from various sources and by various methods. The wettability of nanocellulose is based on the presence of oxygen-containing chemical functional groups (-OH) in its molecules, and can be further modulated by pretreatment of nanocellulose by oxidation during its preparation. This oxidation is usually catalyzed by 2,2,6,6-tetramethylpiperidine-1-oxyl radical (TEMPO), and it endows the nanocellulose with –COOH groups (for a review, see [10]). It is generally known that a moderate wettability of the material surface induce the adsorption of cell adhesion-mediating proteins from biological fluids in an appropriate geometrical conformation, accessible by cell adhesion receptors, and it therefore enhances the cell adhesion and growth (for a review, see [145]). The –COOH groups also endows the nanocellulose with negative charge, which can be of various density according to the content of these groups [61]. The charge density then modulates the morphology and roughness of nanocellulose films [146] and also the interaction of nanocellulose with cells in terms of their growth, viability, and susceptibility for transfection with DNA constructs [61]. The type of charge (i.e., positive or negative) is also important for the interaction of nanocellulose with cells. On the one hand, both types of charge improved the adhesion and growth of cells and reduced cell immune activation in comparison with uncharged nanocellulose [147,148]. On the other hand, anionic nanocellulose often provided a more suitable support for the cell adhesion and growth than cationic nanocellulose, created by modifying the nanocellulose with ammonium groups [149]. However, the anionic nanocellulose elicited a more pronounced immune response of cells than cationic nanocellulose [150]. The role of physicochemical properties of nanocellulose in the cell-material interaction is discussed more deeply in the following parts of this review, particularly in Section 3, Section 4, Section 5 and Section 6.

## 2. History of Nanocellulose Research with Focus on Biomedical Applications

Cellulose in general has been investigated for tens of years (for a review, see [151]). However, nanocellulose has emerged as a promising material in the last 12 years, as indicated by three widely used databases, namely ProQuest, Web of Science (WOS), and PubMed (Figure 1). The largest ProQuest database includes the widest range of publication sources, such as scholarly journals, trade journals, books, dissertations and theses, newspapers, magazines, reports, wire feeds, blogs, podcasts, and websites blogs. The document types indexed by WOS are original and review articles in scientific journals, papers in conference proceedings, books and book chapters, editorial materials, news items, and corrections. The PubMed database, which is relatively small but widely used by biologists and physicians, indexes mainly original and review articles published in impacted journals. Using the search term “nanocellulose,” 5864, 2918, and 683 publications in total were found in the ProQuest, WOS and PubMed databases, respectively, from 2006 to 2018. 

Interestingly, the first paper using the term “nanocellulose,” appearing in June 2006, was a study by Kramer et al. [140], dedicated to the potential use of the nanocellulose as a biomaterial for constructing tissue replacements. This idea was inspired by several favorable properties of nanocellulose, such as water absorption capacity, appropriate strength and elasticity, controllable shape, nanofibrous and porous structure, and biocompatibility. Specifically, the first paper was focused on developing collagen-like materials based of the composites of bacterial cellulose and synthetic polymers, prepared by photopolymerization of acrylate and methacrylate monomers and methacrylate crosslinkers [140]. The second paper published by the same group of authors in August 2006 was a review article dealing with technical and biomedical applications of nanocellulose, such as creating of artificial blood vessels, cuffs for nerve surgery, animal wound dressings, and cosmetic tissues [141]. 

In 2007, an important paper, dedicated to the use of nanocellulose in tissue engineering, was a study by Bodin et al. [152], focused on creating bacterial cellulose nanofibrous scaffolds functionalized with cell adhesion-mediating GRGDS oligopeptides. These scaffolds enhanced the adhesion of human vascular endothelial cells in vitro and were promising for vascular tissue engineering [152]. Other interesting papers were focused on the types of nanocellulose and methods of their preparation [20], and the use of nanocellulose for reinforcing various materials, such as acrylic latex films [24], poly-L-lactic acid and cellulose acetate butyrate [153]. The latter trend continued in 2008, when nanocellulose was applied for reinforcing shape memory polyurethanes [154] and poly(epsilon-caprolactone) [155]. Electrodes modified with thin films composed of sisal-derived nanocellulose and poly-(diallyldimethylammonium chloride) for detection of triclosan, i.e., an antibacterial and antifungal drug, were also developed [82]. Other papers were focused on the types, adhesion properties, and surface properties of nanocellulose [156], on the preparation of nanocellulose from sisal fibers [157], and on the preparation of cellulose nanofibrils with low and high charge density from wood pulp [146]. The nanofibrils of low charge density formed a network structure, while the nanofibrils of high charge density formed denser film structure. At the same time, the average rms roughness of the films was higher for the low charged films [146].

In 2009, the papers dedicated to biomedical application of nanocellulose continued with a review article by Klemm et al. [21], dealing with the types, sources, modes of preparation, and properties of nanocellulose, with focus on bacterial cellulose as a suitable material for wound dressings and body implants, such as replacements of blood vessels and bone tissue [21]. Another review paper by Oksman et al. **[158]** presents a new research field of bionanocomposites, where different types of nanocelluloses are used as reinforcements in biopolymers. Examples of other interesting papers included modification of the inner structure of bacterial nanocellulose, e.g., its pore size, using polyethylene glycol and carbohydrate additives [159], preparation and characterization of nanoscale cellulose films with different crystallinities [160], and preparation of novel xylan films reinforced by nanocellulose whiskers [161].

In 2010, the current and future applications of bacterial nanocellulose in the biomedical field, particularly as biological implants, wound dressings, and scaffolds for tissue regeneration, were further reviewed [162]. Other studies were focused mainly on technical applications of nanocellulose, e.g., on further development of cellulose films, namely on nanocellulose-reinforced methylcellulose-based biodegradable films [163] and on nanocellulose-reinforced chitosan composite films as edible films or coatings that enhance the shelf life of foods [164]. Other examples included studies on safe, effective, and well-controlled production of bacterial nanocellulose [1,165] and on preparation of electrically-conductive nanocellulose-polypyrrole composites applicable for ion-exchange and in paper-based energy storage devices [166].

In 2011 and 2012, the number of studies focused on the use of cellulose in various technological applications, including biotechnology, increased rapidly. For example, nanocellulose aerogels were proposed as oil absorbents for water purification [39], and nanocellulose liners were designed as adsorbents for protecting personnel from chemical and biological hazards [167]. Hybrid materials consisting of bacterial nanocellulose and photocatalytically active TiO_2_ nanoparticles were developed for drinking water purification and air cleaning [168], and nanocellulose-TiO_2_ hybrid films were shown to be promising as transparent coatings, where high wear resistance and UV activity are required [169]. Nanofibrillated cellulose was also investigated as a carrier for titanium dioxide, zinc oxide and aluminum oxide nanotube aerogels for potential application in sensorics, e.g., as humidity sensors [40]. Supercapacitors consisting of bacterial nanocellulose papers, carbon nanotubes, and triblock-copolymer ion gels were proposed for energy storage [115]. Nanocellulose was isolated from sources other than bacterial cellulose, namely from plant materials such as sugar cane bagasse [73], birch pulp [57,58,59] and kraft pulp [170]. In the field of biotechnological applications, nanocellulose paper sheets were used for extracting DNA oligomers [104], and a cellulose-based hydrogel was used for immobilizing trypsin [100]. Cellulose surfaces modified by irreversible adsorption of carboxymethyl cellulose were created as a platform for covalent binding of antibodies for immunoassays, e.g., for detection of hemoglobin [171]. Cellulose nanofibers were tested as a novel tableting material [57], and were also proposed for the construction of films for sustained parenteral delivery of drugs, e.g., analgesics, antiphlogistics, corticoids, and antihypertensives [58,59]. In combination with silver nanoclusters, nanofibrillated cellulose was designed as a novel composite with fluorescence and antibacterial activity for potential wound dressings [172]. In combination with polypyrrole, nanocellulose was proposed for constructing hemodialysis membranes [106]. Original papers in the field of tissue engineering also appeared. Small-diameter vascular grafts were grown from bacterial cellulose around supporting silicone tubes (3 mm or 6 mm in diameter), and then tested in a pig model [173]. Nanocomposites of bacterial cellulose and hydroxyapatite were created for bone healing applications using a biomimetic mineralization in simulated body fluid. These composites supported the attachment of mouse MC3T3-E1 osteoprogenitor cells and their osteogenic differentiation, determined by alkaline phosphatase gene expression [174]. Last but not least, novel injectable scaffolds made of nanofibrillar cellulose hydrogels, derived from birch pulp, were prepared. These scaffolds supported the differentiation of human hepatic cell lines HepaRG and HepG2 in vitro, and maintained the viability of human retinal pigment epithelial ARPE-19 cells after injection with syringe needles of various sizes [175].

In 2013, there was a great burst of studies dealing with the potential use of nanocellulose in tissue engineering, or at least with the interaction of cells with nanocellulose. One of the first original papers published in January 2013 was concentrated on the potential use of nanocellulose in neural tissue engineering [176]. In this study, composite membranes consisting of bacterial nanocellulose (BNC) and polypyrrole (PPy) were used as a template for seeding PC12 rat neuronal cells. These cells adhered and grew significantly better on BNC/PPy composites than on pure BNC. In addition, the presence of electrically conductive PPy made electrical stimulation of cells possible, and this is considered to be beneficial for various cell functions [176]. In another study, tubular structures made of bacterial nanocellulose were successfully applied in rats in vivo as conduits for regeneration of the damaged femoral nerve. These structures prevented excessive proliferation of connective tissue and penetration of the damaged nerve with scar tissue, which is the main obstructive agent for the growth of neurites during nerve regeneration. In addition, cellulosic “neurotubes” allowed the accumulation of neurotrophic factors inside, which further facilitated nerve regeneration [177]. Nanocellulose was also tested for reconstructing human auricle in vitro using bacterial nanocellulose scaffolds [178], combined with primary human chondrocytes obtained during routine septorhinoplasties and otoplasties [179]. Another in vitro model of articular cartilage was bovine knee cartilage with a punch defect filled with bacterial nanocellulose [180]. Bacterial nanocellulose wound dressings were successfully applied for healing large-area and full-thickness skin defects in mice in vivo [181]. Bacterial nanocellulose scaffolds, improved by conjugation with fibronectin and type I collagen, proved to be excellent substrates for the adhesion of human umbilical vein endothelial cells and mouse mesenchymal stem cells of the line C3H10T1/2 [182]. Not only cellulose nanofibrils present in bacterial nanocellulose, but also other forms of nanocellulose, such as nanowhiskers or nanocrystals, were shown to have great potential in tissue engineering and in other biomedical applications [183].

## 3. Recent Use of Nanocellulose in Tissue Engineering and Tissue Repair

In the last five years, i.e., from 2014 to 2018, the use of nanocellulose in tissue engineering and related areas, such as wound healing and cell-material interaction, has been further developed, together with applications of nanocellulose in industry and technology, including various biotechnologies, such as biosensing and controlled drug delivery (Table 2; for a review, see [92,184,185,186,187,188,189]). Research on the potential use of nanocellulose in neural tissue engineering, cartilage tissue engineering and skin wound dressings, and also in liver, vascular and bone tissue engineering, as mentioned above, continued with several promising achievements. 

In neural tissue engineering, it was demonstrated for the first time that SH-SY5Y neuroblastoma cells, cultured on three-dimensional (3D) bacterial nanocellulose (BNC) scaffolds, adhered, proliferated and also differentiated toward mature neurons, as indicated by functional action potentials detected by electrophysiological recordings [190]. The adhesion, proliferation, and formation of 3D neuronal networks on 3D BNC scaffolds can be further enhanced by cationic modification of this material, i.e., on trimethyl ammonium betahydroxy propyl cellulose, as demonstrated on PC12 cells, a widely-used model of neurons [147]. In addition to their potential use in neural tissue replacements, nanocellulose-based neural tissue-engineered constructs were designed as innovative tools for brain studies. For this purpose, an ink that contained wood-derived cellulose nanofibrils and carbon nanotubes was used for 3D printing of electrically conductive scaffolds, which promoted the adhesion, growth and differentiation (manifested by elongation of neurites) of human SH-SY5Y human neuroblastoma cells [191].

In cartilage tissue engineering, the high water-retention capacity and the high mechanical strength of cellulose nanofibrils have led to the further development of applications of bacterial nanocellulose for auricular cartilage reconstruction. It was found that BNC with an increased cellulose content of 17% is a promising non-resorbable biomaterial for auricular cartilage tissue engineering, due to its similarity with auricular cartilage in terms of mechanical strength and host tissue response [2]. Other promising materials for this application were bilayered scaffolds composed of BNC and alginate, which were non-cytototoxic, non-pyrogenic and promoted the growth of human nasoseptal chondrocytes [192]. For articular cartilage engineering, BNC scaffolds modified by laser perforation were used as substrates for the cultivation of human chondrocytes derived from the cartilage covering femoral condyles. These novel scaffolds improved the diffusion of nutrients, the ingrowth and differentiation of chondrocytes, and the deposition of their newly synthesized extracellular matrix within the scaffolds [49]. A further novelty was the application of nanocellulose-based bioink in 3D bioprinting with living cells. A bioink consisting of wood-derived nanofibrillated cellulose and alginate—and containing human articular chondrocytes—was used for 3D printing of anatomically shaped cartilage structures, such as a human ear and sheep meniscus [193]. A similar bioink was used for 3D printing together with irradiated human chondrocytes and induced pluripotent stem cells (iPSC), both derived from articular cartilage [194]. An alginate sulfate/BNC bioink promoted spreading, proliferation, and collagen II synthesis in bovine chondrocytes from femoral condyle cartilage [195]. Another interesting composite material developed for cartilage tissue engineering was a double cross-linked interpenetrating polymer network of sodium alginate and gelatin hydrogels, reinforced with 50 wt % of cellulose nanocrystals [196]. Nanocellulose is also promising for the treatment of intervertebral disc degeneration. Gellan gum hydrogels reinforced with cellulose nanocrystals were designed as substrates for regenerating the annulus fibrosus, i.e., the outer part of the discs [197].

From 2014 to 2018, nanocellulose has been increasingly applied in other interesting areas of experimental tissue engineering, namely in liver tissue engineering, adipose tissue engineering, vascular tissue engineering, bone tissue engineering and bone implant coating, and in reconstruction of the urethra and the *dura mater*.

In liver tissue engineering, the first idea was to create a 3D culture of hepatic cells, which is more physiologically relevant than the 2D culture that is traditionally used to predict and estimate the metabolism, excretion and toxicity of drugs and other chemicals in the human liver. For this purpose, 3D scaffolds based on birchwood-derived nanofibrillar cellulose were generated. These scaffolds promoted differentiation and proper functioning of human liver progenitor cells of the line HepaRG, derived from a liver tumor of a female patient who was suffering from a hepatitis C virus infection and hepatocarcinoma. Specifically, the HepaRG cells formed 3D multicellular spheroids with apicobasal polarity and functional bile canaliculi-like structures. In addition, hepatobiliary drug transporters, i.e., MRP2 and MDR1, were localized on the canalicular membranes of the spheroids, and vectorial transport of fluorescent probes toward the biliary compartment was demonstrated. Cell culture in a 3D hydrogel supported the mRNA expression of hepatocyte markers (albumin and CYP3A4), and the metabolic activity of CYP3A4 in the HepaRG cell cultures [198].

Similarly, in adipose tissue engineering, efforts were made to create a 3D in vitro model of adipose tissue for studies on adipose biology and on metabolic diseases, such as obesity and diabetes. For this purpose, 3D scaffolds were prepared by crosslinking homogenized bacterial nanocellulose fibrils using alginate and by freeze-drying the mixture to obtain a porous structure. When seeded with mesenchymal stem cells of the line C3H10T1/2, derived from mouse embryos and incubated in an adipogenic medium, the 3D scaffolds contained more cells with markers of adipogenic cell differentiation, i.e., growing in clusters and containing large lipid droplets, than 2D bacterial nanocellulose scaffolds. 3D scaffolds therefore have great potential not only for in vitro studies, but also for adipose tissue engineering, for reconstructive surgery after trauma, tumor removal or congenital defects [199]. A similar system was created in a study by Henriksson et al. [200] by 3D printing with the use of a bioink made of nanocellulose and hyaluronic acid, and containing adipocytes. The adipocytes showed uniform distribution throughout the scaffolds, high viability and more mature phenotype than the cells in conventional 2D culture systems.

The 3D environment is also suitable for culture of human pluripotent stem cells (hPSCs) in order to maintain their pluripotency for various biomedical applications, such as drug research and regenerative medicine. A flexible 3D environment for hPSC culture, mimicking the 3D in vivo stem cell niche, was created using a plant-derived nanofibrillar cellulose hydrogel. This hydrogel maintained the pluripotency of hPSCs for 26 days, as evidenced by the expression of transcription factors OCT4 and NANOG, stage specific embryonic antigen-4 (SSEA-4), and also by in vitro embryoid body formation and in vivo teratoma formation [201]. 

For vascular tissue engineering, tubular structures were created from BNC using silicone tubes as molds. These tubes were also considered to have great potential for substituting other hollow organs, including the ureter and the esophagus [202]. In a study by Weber et al. [203], BNC tubes were used to replace the right carotid artery in sheep in vivo. After explantation, a histologic analysis revealed no acute signs of foreign body reaction, such as immigration of giant cells or some other acute inflammatory reaction, and therefore provided evidence for good biocompatibility of the tubes. However, the tubes were highly prone to thrombotic occlusion, and their implantation required antiplatelet therapy [203]. Another interesting idea was to use bacterial nanocellulose coupled with superparamagnetic iron oxide nanoparticles for coating endovascular stents, which will then attract vascular smooth muscle cells (VSMCs) for in situ reconstruction of the *tunica media* in blood vessels. In experiments in vitro, magnetic BNC coated with polyethylene glycol proved to form suitable scaffolds for porcine VSMCs, showing minimum cytotoxicity and supportive effects on cell viability and migration. This material also possessed suitable mechanical properties, and was considered to be promising for the treatment of brain vascular aneurysms [204,205]. Nanocellulose scaffolds were also applied for studies on vasculogenesis. BNC scaffolds functionalized with IKVAV peptide, i.e., a laminin-derived ligand for integrin adhesion receptors on cells, were used for studies on vasculogenic mimicry of human melanoma SK-MEL-28 cells, and appeared to provide a promising 3D platform for screening antitumor drugs [50].

BNC, even in its unmodified state, also showed a great promise for bone tissue engineering. BNC without additives stimulated the adhesion, multilayered growth and osteogenic differentiation of bone marrow mesenchymal stem cells (MSCs) derived from rat femur. As revealed by Second Harmonic Generation (SHG) imaging, the MSCs on BNC scaffolds produced a mature type of collagen I and showed activity of alkaline phosphatase [206]. Wood-derived nanofibrillated cellulose is also promising for the construction of scaffolds for bone tissue engineering, as proved on human MSCs grown on composite scaffolds containing this cellulose and chitin [207].

The performance of MSCs and other bone-forming cells, e.g., rat calvarial osteoblasts, on nanocellulose-based scaffolds can be further improved by biomimetic mineralization with calcium phosphates, such as hydroxyapatite and tricalcium phosphate [7,208,209]. In addition, these scaffolds can be coupled with collagen I or with osteogenic growth peptide [52]. Nanocellulose is also promising for bone implant coating. A hybrid coating, consisting of 45S5 bioactive glass individually wrapped and interconnected with fibrous cellulose nanocrystals (CNCs), was deposited on 316L stainless steel in order to strengthen bone-to-implant contact and to accelerate the bone healing process. This coating substantially accelerated the attachment, spreading, proliferation and differentiation of mouse MC3T3-E1 osteoblast progenitor cells in vitro and mineralization of the extracellular matrix deposited by these cells [210]. Similarly, coating 3D-printed polycaprolactone scaffolds with wood-derived hydrophilic cellulose nanofibrils enhanced the attachment, proliferation and osteogenic differentiation of human bone marrow-derived mesenchymal stem cells [35]. 

Urethral reconstruction was performed in a rabbit model using 3D porous bacterial cellulose scaffolds seeded with rabbit lingual keratinocytes [211], and in a dog model using smart bilayer scaffolds comprising a nanoporous network of bacterial cellulose and a microporous network of silk fibroin [212]. The bilayer scaffolds were pre-seeded with keratinocytes and smooth muscle cells isolated from dog lingual tissue obtained by biopsy. The nanoporous network provided good support for epithelial cells, while the microporous scaffolds supported the growth and penetration of smooth muscle cells [212].

For reconstruction of the *dura mater*, bacterial cellulose membranes were tested as potential dural patches to prevent leakage of cerebrospinal fluid, which is a common complication after cranial and spinal surgery. These membranes supported the attachment and the viability of human dural fibroblasts [213]. 

Other interesting applications of nanocellulose have included connective tissue repair, repair of congenital heart defects, ophthalmologic applications, creation of protective barriers, and cell transfection. 

For connective tissue repair, softwood pulp–derived cellulose nanocrystals were injected into skin and tendon specimens, isolated from pigs and stretch-injured using a mechanical testing machine. This treatment mechanically reinforced these matrices, which was manifested by the increased elastic moduli and yield strength of the matrices. At the same time, the cellulose nanoparticles showed no cytotoxicity for rat primary patella tendon fibroblasts, as revealed by a WST-1 assay of the activity of mitochondrial enzymes. Moreover, the activity of these enzymes in cells cultivated for 2–3 weeks in the presence of cellulose nanocrystals was significantly higher than in the control untreated cells [62].

For the repair of congenital heart defects, bacterial nanocellulose was used as a new patch material for closing ventricular septal defects in a pig model. This material could serve as an alternative to materials currently used in clinical practice, namely polyester, expanded polytetrafluoroethylene (ePTFE) and autologous or bovine pericardium, which are often associated with compliance mismatch and with a chronic inflammatory response [214].

Ophthalmologic applications of nanocellulose include the construction of contact lenses. For their construction, a highly transparent macroporous hydrogel was developed, consisting of poly (vinyl alcohol) reinforced with cellulose nanofibrils and containing more than 90% of water. The hydrogel exhibited high transparency with a refractive index close to that of water, very good UV-blocking properties and elastic collagen-like mechanical behavior typical for soft tissues [215]. An attempt was also made to reconstruct experimentally damaged cornea in rabbits using pure bacterial cellulose membranes and composite bacterial cellulose/polycaprolactone membranes, but the results were considered as unsatisfactory. Histological examination showed absence of epithelium on the membranes, fibroplasia close to the implants, lymph inflammatory infiltrate with giant cells, collagen disorganization, and the presence of immature collagen fibers [216].

Creating protective barriers involves designing materials that prevent intraperitoneal adhesions or immune rejection of transplanted cells. For example, in experimental abdominal defects in dogs, which were repaired using BNC membranes, negligible intraperitoneal adhesions were detected between the BNC and the intestinal loops in comparison with conventionally used polypropylene meshes [55]. Modifying polypropylene meshes, and also metallic meshes, with BNC enhanced their potential applicability in hernioplasty and cranioplasty [217]. For immunoprotection of transplanted cells, a composite hydrogel consisting of TEMPO-oxidized bacterial cellulose and sodium alginate was developed for encapsulation of cells, e.g., insulin-secreting β-cells of Langerhans islets [218]. A sophisticated nanocomposite membrane was developed for encapsulation of PC12 cells. One of the surfaces of bacterial cellulose (BC) pellicles was coated with collagen to enhance cell adhesion, and the opposite side of the BC pellicles was coated with alginate to protect the transplanted cells from immune rejection. The nanocomposite membrane was permeable to small molecules, i.e., dopamine secreted by the cells, but was impermeable to large molecules, such as IgG antibodies [219]. 

An interesting finding was that nanocellulose can also modulate the efficiency of cell transfection by its structure and electrical charge density. Nanofibrillated cellulose was prepared from birch kraft pulp in the form of films or hydrogels with low or high charge density. The films with low charge density showed a more pronounced increase in the efficiency of transfection of HeLa cells (i.e., a cell line derived from human cervical cancer cells) with DNA constructs, encoding the Red Fluorescent Protein, than the films with high charge density and hydrogels with both low and high charge densities. In addition, matrices with low charge density facilitated the encapsulated HeLa cells and Jurkat cells (i.e., an immortalized line of human T lymphocytes) for ingrowth, survival, and proliferation [61].

The following part of this review is focused on the use of nanocellulose for skin tissue engineering and wound healing.

## 4. Nanocellulose in Skin Tissue Engineering

### 4.1. Bacterial Nanocellulose in Skin Tissue Engineering

Skin tissue engineering involves reconstructing two main layers of the skin, namely the epidermis, i.e., the superficial skin layer formed mainly by keratinocytes, and the dermis, i.e., the skin inner layer formed mainly by fibroblasts. Due to its certain resemblance to natural soft tissues, including skin, bacterial cellulose is the most widely used type of nanocellulose for reconstructing these layers [138]. In fact, bacterial cellulose is a hydrogel containing nanofibrils, which mimics the fibrillar component of natural extracellular matrix. Bacterial cellulose has a great capacity to retain moisture, and it also has appropriate mechanical properties, such as strength, Young’s modulus, elasticity and conformability [14,29,142]. The use of bacterial cellulose in skin reconstruction started long before the first appearance of the word “nanocellulose” in the ProQuest, WOS, and PubMed databases. It was simply called “bacterial cellulose,” though it is a hydrogel containing cellulose nanofibrils. The first report of the use of bacterial cellulose in skin wound therapy came from 1990, when bacterial cellulose pellicles were proposed as “temporary skin substitutes” for treating burns, ulcers, abrasions, and other skin injuries [220]. In 2006, thin films of bacterial cellulose were used as substrates for the cultivation of human transformed skin keratinocyte and human normal skin fibroblast cell lines. The films supported spreading, growth and migration in keratinocytes but not in fibroblasts, which formed clusters and detached from the films. This phenomenon was explained by relatively weak cell-material adhesion in comparison with the relatively strong cell–cell adhesion in fibroblasts, which generates a contractile force [221]. However, in a study by Kingkaew et al. [222], bacterial cellulose films proved to be good substrates for the adhesion, spreading and growth of both human skin keratinocytes and fibroblasts. Similarly, a surface-structured 3D network of bacterial cellulose nanofibers also provided good support for human keratinocytes and fibroblasts and stimulated the healing of experimental skin wounds in mice [13].

The adhesion and growth of skin cells on bacterial cellulose can be further improved by combining it with other biologically active molecules. For example, the adhesion of human keratinocytes on bacterial cellulose films was improved by enriching these films with chitosan [222]. Incorporation of keratin, isolated from human hair, into bacterial cellulose improved the attachment, proliferation and morphology of human skin keratinocytes of the HS2 cell line and human skin fibroblasts of the Detroit 562 cell line [223]. Composite scaffolds made of microporous regenerated bacterial cellulose and gelatin provided good support for the adhesion and proliferation of human keratinocytes of the HaCaT line, and for their penetration into the scaffolds (to a depth of 300 μm). In experiments *in vivo* performed in mice, scaffolds with gelatin showed greater wound closure efficacy (93%) than pure bacterial cellulose (63%) [224]. Electroactive composites of bacterial cellulose and conducting polymers, such as polypyrrole and polyaniline, also hold promise for skin tissue engineering [225].

The potential use of bacterial cellulose in skin regeneration and in other areas of tissue engineering has been reviewed by many authors, e.g., [12,45,48,142,185,226,227,228]. Novel microporous 3D scaffolds with controllable pore size were prepared from bacterial nanocellulose using paraffin microspheres. These scaffolds supported the proliferation of mouse embryonic NIH 3T3 fibroblasts and were considered to be promising for soft tissue engineering [42]. 

### 4.2. Plant- and Algae-Derived Nanocellulose in Skin Tissue Engineering

Like bacterial nanocellulose, plant-derived nanocellulose has repeatedly been shown to be promising for skin tissue engineering, especially after its physical and chemical properties have been modified. For example, cellulose nanofibrils (CNFs) were modified either by introducing a negative electrical charge using TEMPO-mediated oxidation or by introducing a positive charge using 2,3-epoxy propyl trimethyl ammonium chloride (EPTMAC) [149,229,230]. In a study by Skogberg et al. [149], unmodified (u-), anionic (a-), and cationic (c-) CNFs, derived from hardwood kraft pulp (u-, c-CNF) or from softwood kraft pulp (a-CNF) were fabricated using an evaporation-induced droplet-casting method on glass. Atomic force microscopy showed a significantly higher degree of orientation of nanofibers along a single line on c- and u-CNF surfaces than on a-CNF surfaces. Both a-CNF and c-CNF surfaces supported the adhesion, spreading, viability and proliferation of mouse embryonic fibroblasts, though the cell performance was better on a-CNF. However, the cells on aligned c-CNF surfaces showed orientation in parallel, which could be utilized for guided cell growth. Recently, transferrable free-standing nanocellulose films have also been produced with a similar alignment of CNFs in parallel to an evaporating liquid boundary line during evaporation [231]. When an electrical charge is introduced into nanocellulose, it can be functionalized with various biomolecules, e.g., cell adhesion peptides [229] and silk fibroin [230], which improves the capacity of nanocellulose for colonization with cells and for wound healing. 

In our recent experiments in collaboration with Skogberg and her colleagues, human dermal fibroblasts were cultured on cellulose meshes in a DMEM medium with 10% of fetal bovine serum and 40 μg/ml of gentamycin. Two different types of nanocellulose solutions, i.e., c-CNF, a-CNF, were applied on the surface of a cellulose mesh (obtained from Holzbecher Ltd., Zlic, Czech Republic) in order to cover its microfibrous structure. Both c-CNF and a-CNF were expected to improve the surface properties of the cellulose mesh for cell adhesion and proliferation. A 0.15 wt % c-CNF solution formed a thin film on the surface of the cellulose meshes, while the 0.15 wt % a-CNF solution covered individual cellulose microfibers and filled the wide spaces between them. This may be due to a lower degree of fibrillation of c-CNF in comparison with a-CNF, which results in a solution with larger fibers in the case of c-CNF. Larger c-CNF fibers cannot penetrate into the pores of the cellulose mesh. They cumulate on the top of the mesh and form a film-like structure there. However, the smaller a-CNF fibers can leak in to the pores of the cellulose mesh. Our results have shown positive effects of both types of CNF coverings on the adhesion and proliferation of dermal fibroblasts. However, we observed that a-CNF was more suitable for adhesion and growth of dermal fibroblasts than c-CNF, on which the cells were often round, less spread and proliferated relatively slowly. The morphology of human dermal fibroblasts was more physiological on a-CNF than on c-CNF. The cells on a-CNF adhered along the cellulose fibers, spread between them and formed a better-developed filamentous actin (F-actin) cytoskeleton (Figure 2).

Other chemical modifications of plant nanocellulose intended for skin tissue engineering include converting it to cellulose acetate or to hydroxyethyl cellulose. Conversion to cellulose acetate is known to enhance the electrospinnability of cellulose, as was demonstrated in cellulose extracted from sugar cane bagasse, and the electrospun fibrous scaffolds then supported the adhesion and growth of mouse subcutaneous fibroblasts of the line L929. The cell behavior was further improved by blending the cellulose with poly (l-lactide) or with polydioxanone [74]. Three-dimensional cellulose acetate scaffolds, produced by an electrohydrodynamic direct jet process called spin-printing, stimulated the adhesion and metabolic activity of human dermal fibroblasts to a greater extent than polycaprolactone scaffolds with a similar fibrous morphology and pore geometry [232]. Blending cellulose acetate with gelatin can modulate its applicability for skin tissue engineering or for wound dressing. The scaffolds with a lower content of cellulose acetate and a higher content of gelatin (ratio 25:75) promoted high proliferation activity of human dermal fibroblasts and adhered to a wound, showing that they were promising for skin tissue engineering. By contrast, the scaffolds with a higher content of cellulose acetate and a lower content of gelatin (ratio 75:25) appeared to be suitable for low-adherent wound dressings [233]. Other cellulose acetate-based scaffolds with potential for skin tissue engineering include composite 3D electrospun cellulose acetate/pullulan scaffolds, which promoted the adhesion and growth of mouse L929 fibroblasts [234], and composite biomimetic nanofibrous gelatin/cellulose acetate/elastin scaffolds, which promoted the adhesion and growth of human gingival fibroblasts [235]. Nanofibrous scaffolds prepared by rotary jet spinning from cellulose acetate and soy protein hydrolysate are another promising material. In vitro, these scaffolds promoted the migration and proliferation of dermal fibroblasts, their infiltration inside the scaffolds and their expression of β_1_-integrin adhesion receptors. In vivo, these scaffolds accelerated re-epithelialization and epidermal thinning, and also reduced scar formation and collagen anisotropy [236].

Hydroxyethyl cellulose is another modification of cellulose that can be used for creating nanostructures. This modification of cellulose is water-soluble and, like cellulose acetate, it can be used for electrospinning of nanofibrous scaffolds. Nanofibrous scaffolds made of hydroxyethyl cellulose blended with poly(vinyl alcohol) supported the adhesion and growth of human skin fibroblasts [237]. The behaviour of the fibroblasts was further improved by adding collagen into the blend, and the antimicrobial activity of the scaffolds was established by adding silver nanoparticles without a considerable increase in the cytotoxicity of the scaffold for the fibroblasts [238].

Plant-derived nanocellulose in the form of nanocrystals can be used advantageously for reinforcing materials typically used for tissue engineering, such as degradable natural and synthetic polymers, which are relatively weak. Cellulose nanocrystals (CNCs) are produced by acid hydrolysis of cellulose fibers, employing either sulfuric acid or hydrochloric acid. Due to their structural defects, CNCs have a very large elasticity modulus (about 130 GPa), which is similar to that of Kevlar, and they have high strength (about 7 GPa). In addition, CNCs have low extension to break, high aspect ratios, high surface areas, high crystallinity, and apparent biocompatibility [16,78]. CNCs were used to reinforce collagen films, and these composites, also supporting the viability of mouse embryonic 3T3 fibroblasts, were promising for skin tissue engineering [16]. In another study, cotton-derived cellulose nanocrystals were electrospun together with poly (lactic-*co*-glycolic acid) (PLGA). The resulting scaffolds improved the adhesion, spreading, and proliferation of 3T3 fibroblasts in comparison with neat PLGA nanofiber membranes [239]. 

Nanocellulose derived from *Cladophora* algae can also be improved for tissue engineering purposes by physicochemical modifications. The adhesion and spreading of human dermal fibroblasts were relatively poor on unmodified *Cladophora* nanocellulose films, but they increased on nanocellulose carboxylated by electrochemical TEMPO-mediated oxidation. This increase was proportional to the degree of oxidation of the material [84].

### 4.3. Limitations of the Use of Nanocellulose in Skin Tissue Engineering

In spite of all the encouraging results mentioned above, the use of nanocellulose (and cellulose in general) in skin tissue engineering is limited by its non-degradability in the human organism. The retention of non-degradable material in skin could induce scar formation. Degradability of cellulose can be induced by incorporating cellulase enzymes, as demonstrated in bacterial cellulose [240], especially in conjunction of these enzymes with β-glucosidase [241]. Degradable cellulose can also be created by introducing N-acetylglucosamine residues into the cellulose molecule during its synthesis by metabolically engineered *Gluconacetobacter xylinus*. These residues then render the cellulose molecules susceptible to degradation by lysozyme, an enzyme that is widespread in the human body [242,243]. Another approach for rendering cellulose-based scaffolds degradable, at least partially, is oxidation. Oxidized acetyl cellulose sponges, implanted subcutaneously into rats in vivo, showed degradation of 47% of their dry mass after 60 weeks, while in ethyl cellulose the proportion was only 18% [244]. Regenerated cellulose (methylolcellulose) and 2,3-dialdehydecellulose (DAC) have also been considered as degradable, although only at a slow rate. In addition, DAC membranes supported adhesion, growth and extracellular matrix formation in human neonatal skin fibroblasts cultured on these materials [245]. Last but not least, cellulose derived from *Styela clava* tunics is also slowly degradable. After subcutaneous implantation into rats for 90 days, the weight loss was greater in cellulose films from *Styela clava* (almost 24% of their initial weight) than in films prepared from wood pulp cellulose (less than 10%) [89]. 

### 4.4. Nanocellulose as a Carrier for Cell Delivery into Skin Defects 

Although there has been only limited direct use of cellulosic materials, including nanocellulose, in skin tissue engineering, these materials, even in their non-degradable forms, can be used indirectly for skin tissue engineering, i.e., as carriers for delivering cells into wounds. After the cells have adhered to the wound bed, they can be released from the scaffolds, and the scaffolds can be removed. Experiments in vitro, performed on a bacterial cellulose/acrylic acid (BC/AA) hydrogel colonized by epidermal keratinocytes (EK) and dermal fibroblasts (DF), showed that from day 1 to day 3 after seeding on BC/AA, about 63% of EK and 69% of DF were cumulatively transferred from BC/AA on to an ovine collagen hydrogel [138]. Experiments in vivo performed in mice showed that BC/AA hydrogels loaded with cells produced the greatest acceleration on burn wound healing, followed by treatment with hydrogel alone, and by the untreated group. On day 13 after wound coverage, the percentage of wound reduction for the hydrogel loaded with cells, for the pure hydrogel and for the control untreated group of animals were about 77%, 72%, and 65%, respectively. The transferred cells are believed to assist in initiating the wound healing process, where the fibroblasts play a role in forming the granulation tissue and the keratinocytes help in re-epithelialization [246]. Wound healing can also be accelerated by transferring mesenchymal stem cells, seeded on nanocellulose-based carriers, into the damaged skin. For example, membranes of bacterial cellulose with gellan gum, incorporated with the antifungal drug fluconazole, were developed for delivery of human adipose-derived mesenchymal stem cells (ASCs), obtained by liposuction. The membranes with ASCs were applied for covering second-degree burn wounds produced in rats. Fluorescence staining with FITC and DAPI proved that the ASCs were transferred into the wounds. The transferred ASCs can improve wound healing directly, by proliferating and differentiating in the host tissue, and mainly indirectly, by their paracrine secretion of a wide range of bioactive molecules, such as cell-adhesion mediating molecules, immunomodulatory molecules, growth factors, and angiogenic factors [247]. Carboxymethyl cellulose (CMC) combined with rat ASCs, obtained from visceral fat, was tested for treating skin lesions created by punch in a dorsal region of rats. In this model, commercially available sodium CMC at a concentration of 10 mg per 1 ml of the culture medium associated with ASCs, increased the rate of cell proliferation of the granulation tissue and the epithelium thickness in comparison with untreated lesions but did not increase the collagen fibers or alter the overall speed of wound closure. In addition, the use of CMC was safe up to a concentration of 20 mg/ml. At a higher concentration of 40 mg/ml, the sodium CMC showed a certain genotoxicity, although this was small and transient, as revealed by an alkaline comet assay [139]. Other cellulose-based carriers for human ASCs were threads prepared from nanofibrillated cellulose, extracted from plants and cross-linked with glutaraldehyde. Cross-linked threads were not cytotoxic for ASCs and supported their adhesion, migration and proliferation in vitro. After intradermal suturing with ASC-decorated threads in an ex vivo experiment performed on porcine skin, the ASCs remained attached to the multifilament sutures without displaying morphological changes or reducing their metabolic activity [248]. In our recent study, novel cell carriers based on clinically used CMC fabrics (Hcel® NaT), modified with fibrin nanofibers, were designed for potential delivery of dermal fibroblasts into skin wounds [249].

## 5. Nanocellulose in Wound Healing

### 5.1. Bacterial Nanocellulose 

#### 5.1.1. Bacterial Nanocellulose without Additives

Similarly as in skin tissue engineering, bacterial nanocellulose (BNC) is considered to be one of the most suitable materials for wound dressing. This is due to its favorable physical, chemical and biological properties, as mentioned above, such as chemical purity, favorable mechanical properties, and water-absorbing capacity [14,29,56,142]. BNC itself, i.e., without any additives, showed a great capacity to stimulate wound healing, i.e., regeneration of the epidermis and dermis. For example, as mentioned above, BNC wound dressings improved the healing of full-thickness skin defects of a relatively large area (2 × 2 cm), created surgically on the back in mice in vivo, in comparison with the control untreated mice. At the same time, BNC-treated mice showed a lower inflammatory response, evaluated by the amount of neutrophils, lymphocytes and macrophages in histological sections. In addition, a cytotoxicity test, performed in vitro on NIH/3T3 fibroblasts, demonstrated that the growth rate of the cells seeded on BNC films was more than 80% of the value obtained in cells grown in standard culture wells, which indicated low cytotoxicity of BNC [45]. Similar results were obtained when bacterial cellulose membranes were applied for 15 days on second-degree burn wounds (1 × 1 cm) produced by contact with a heated metal plate. Bacterial cellulose accelerated the process of healing in comparison with a conventionally used gauze, as manifested by greater thickness of the regenerated epidermis and dermis, a higher number of newly-created blood vessels, a higher level of collagen expression, and a lower number of mast cells infiltrating the damaged site. At the same time, bacterial cellulose did not show toxic effects on the liver and kidney, as revealed by the levels of alanine transaminase, aspartate transaminase, alkaline phosphatase, blood urea nitrogen, creatine and lactate dehydrogenase in the blood serum [250]. A recent study by Kaminagakura et al. [54] showed that bacterial cellulose membranes (Nanoskin^®^, Innovatecs Biotechnological Products Ltda, ME, São Carlos-SP, Brazil ) promoted healing of full-thickness skin wounds in guinea pigs, created by surgical removal of skin from their dorsal region (2 × 4 cm), to a similar extent as in the control autologous skin implants. A coating of Nanoskin^®^ with gelatin did not further improve the healing effect. However, skin wound healing can be modulated by the architecture of bacterial cellulose films. The bottom side of these films was constructed with a larger pore size, and with a looser and rougher structure than that of the top side. A microfluidics-based in vitro wound healing model revealed that the bottom side of the films better promoted the migration of cells to facilitate wound healing. These scaffolds are therefore also promising for skin tissue engineering. Moreover, full-thickness skin wounds in Wistar rats, covered by the bottom side of the films, showed faster recovery and less inflammatory response than the top side of these films and the traditionally-used gauze [251]. Finally, an interesting application of bacterial nanocellulose was for creating transparent wound coverings, which allowed optical real-time monitoring of wound healing, and also diagnostics of the infection and inflammation in chronic wounds [252]. Another transparent wound dressing was developed by combining bacterial cellulose whiskers with a poly (2-hydroxyethyl methacrylate) hydrogel and silver nanoparticles, which endowed the dressing with antibacterial activity. At the same time, this material facilitated the growth of NIH 3T3 fibroblasts, which indicated its non-toxicity [31].

#### 5.1.2. Bacterial Nanocellulose with Additives

In order to further improve the healing effect of bacterial (nano) cellulose, this material has been combined with other biologically-active molecules, such as other polysaccharides, proteins, glycosides, cytokines and growth factors, local anesthetics, and even nanoparticles. For example, combination with chitosan improved the mechanical properties and endowed bacterial cellulose-based wound dressings with antimicrobial properties [253]. The mechanical properties of composite electrospun nanofibrous mats containing bacterial nanocellulose and chitosan were further improved by adding medical grade diamond nanoparticles to the electrospinning solution. Introducing these nanoparticles facilitated the electrospinning process and reduced the diameter of the fibers. Moreover, the nanodiamond-modified mats were more hydrophilic and thus more attractive for the adhesion and growth of mouse skin L929 fibroblasts, which made them promising for skin tissue engineering [29]. 

An important protein for modifying bacterial cellulose is sericin, which is created by silkworms (*Bombyx mori*) as a component of silk. A silk sericin-releasing bacterial nanocellulose gel was developed to be applied as a bioactive mask for facial treatment [254]. Silk sericin diffusing from bacterial cellulose did not influence the growth of keratinocytes but enhanced the proliferation of fibroblasts, increased the cell viability and improved the production of extracellular matrix. Bacterial cellulose/silk sericin composites are therefore promising not only for wound dressing applications but also for tissue engineering [255]. A bacterial nanocellulose wound dressing with sericin and polyhexamethylene biguanide (PHMB), an antimicrobial agent, was clinically tested in volunteers [133]. 

An important cytokine used for bacterial cellulose modification was macrophage-stimulating protein (MSP), a cytokine highly expressed in ASCs and probably playing a critical role in wound healing. In an in vivo study, MSP was applied to a full-thickness skin wound with bacterial cellulose membranes, and this treatment accelerated the wound healing, probably by migration of dermal fibroblasts, which have receptors for MSP, and by enhanced synthesis and remodeling of collagen [256]. Smart membranes made of oxidized bacterial cellulose incorporated with epidermal growth factor (EGF) were developed in order to enhance the process of re-epithelialization. The release of EGF was triggered by lysozyme, an enzyme commonly found at infected skin wounds [257]. Re-epithelialization of skin wounds in rats was also enhanced by bacterial cellulose membranes incorporated with vaccarin, a flavonoid glycoside known to promote neovascularisation [47]. In the field of local anesthetics, lidocaine was incorporated into bacterial cellulose in order to reduce pain, especially in burn wounds, and thus to improve the wound healing [132]. Another system developed for lidocaine delivery was based on biodegradable microneedles manufactured from bacterial cellulose and fish scale–derived collagen [36]. A further useful modification of bacterial cellulose is the introduction of glycerin. Glycerin has a characteristic moisturizing effect, which could be clinically relevant for the treatment for skin diseases accompanied by dryness, such as psoriasis and atopic dermatitis [258]. Bacterial nanocellulose usually occurs in the form of nanofibrils, but nanocrystals have also been prepared from this material. The bacterial cellulose nanocrystals were then used to reinforce regenerated chitin fibers, and these composite fibers are applicable for suturing skin wounds [259]. 

An important issue is that wound dressings should protect the wound from microbial infections, which are caused mainly by bacteria. Although bacterial nanocellulose is considered to be an almost ideal wound dressing, it exhibits no antibacterial properties when used by itself. Therefore, numerous studies have dealt with incorporating bacterial cellulose with various antibacterial agents, such as metal-based agents, antiseptics, antibiotics and various nature-derived antibacterial molecules. Metal-based agents used for bacterial cellulose modification include various forms of silver, such as silver sulfadiazine [260] and silver nanoparticles [261], both of which are active against *Pseudomonas aeruginosa*, *Escherichia coli*, and *Staphylococcus aureus*. Silver nanoparticles were further combined with magnetic Fe_3_O_4_ nanoparticles in order to increase the wound healing efficiency of bacterial nanocellulose [262]. Other metal-based nanoparticles with activity against Gram-negative bacteria, combined with bacterial cellulose, were 4,6-diamino-2-pyrimidinethiol (DAPT)-modified gold nanoparticles [263]. Antiseptics used in bacterial cellulose-based wound dressings included povidone-iodine and polyhexamethylene biguanide (PHMB; [56,133]), and also octenidine [264]. Prolonged release of octenidine was achieved by incorporating it into Poloxamer micelles, which were introduced into bacterial nanocellulose [134]. Other antimicrobial drugs incorporated into bacterial cellulose were antimicrobial quaternary ammonium compounds based on fatty acids and amino acids ([EDA][DLA-Tyr]), which were active against *Staphylococcus* aureus and *Staphylococcus epidermidis* [265]. A representative of antibiotics is ceftriaxone, a third-generation cephalosporin [51]. Nature-derived antibacterial molecules used for modifying bacterial cellulose include chitosan, which exhibited bacteriostatic properties against *Escherichia coli* and *Staphylococcus aureus* [253,266]. Other antimicrobial compounds are bromelain, a protease present in pineapple tissues, which also has anti-inflammatory and anticancer properties [267], lignin and lignin-derived compounds [268], and particularly curcumin, i.e., a naturally occurring polyphenolic compound isolated from *Curcuma longa*. The application of curcumin is limited by its extremely low water solubility, which leads to its poor bioavailability. For wound dressings, curcumin was applied mainly with plant-derived and chemically modified nanocellulose, and, in rare cases, in combination with bacterial cellulose. In a recent study, curcumin was entrapped into a composite containing gelatin and ionically modified self-assembled bacterial cellulose and showed wound healing activity and antimicrobial activity [269]. 

In our experiments, we prepared bacterial cellulose loaded with pristine curcumin or with curcumin degradation products. As was mentioned above, pristine curcumin is almost insoluble in polar solvents. In addition, curcumin is unstable in neutral and alkaline pH, and it degrades mainly to ferulic acid, feruloylmethane and vanillin [270]. The degradation of curcumin can also be modulated by temperature. It is known that curcumin starts to degrade at a temperature of approx. 180 °C [271]. In our experiments, degradation products of curcumin were therefore prepared by thermal decomposition of curcumin molecules at temperatures of 180 °C and 300 °C. Fourier-transform infrared spectroscopy (FTIR) and high-performance liquid chromatography (HPLC) detected vanillin and feruloylmethane as the major product at 180 °C, and feruloylmethane at 300 °C. 

Our results showed that bacterial cellulose loaded with curcumin, and particularly with its degradation products obtained at 180 °C, reduced the number of growing colonies of *Staphylococcus epidermis*. Antibacterial activity against *Escherichia coli* was obtained only in samples loaded with the degradation products of curcumin obtained at 180 °C [272].

In vitro tests performed on human dermal fibroblasts revealed that curcumin degraded at 180 °C showed a significant cytotoxic effect on these cells, while curcumin degraded at 300 °C supported the adhesion, spreading and growth of these cells (Figure 3). It therefore appears that vanillin—as the major degradation product at 180 °C—is cytotoxic, and feruloylmethane—as the major degradation product at 300 °C—is non-cytotoxic. However, the antimicrobial and cytotoxic effect of curcumin-loaded bacterial cellulose is strongly dependent on the concentration of curcumin or its degradation products. We observed no cytotoxic effect on fibroblasts at very low doses of curcumin degraded at 180 °C, incorporated into cellulose. 

Other drugs that can be incorporated into bacterial cellulose are anticancer drugs, such as α-mangostin, an antioxidant and antigenotoxic agent derived from the mangosteen tree, which suppressed the growth of B16F10 melanoma cells and MCF-7 breast cancer cells [273]. Bacterial cellulose can also be used for transdermal drug delivery, such as systemic delivery of propranolol, a non-selective β-adrenergic receptor antagonist [131], or diclophenac, a non-steroidal anti-inflammatory drug [130]. Another interesting application of bacterial nanocellulose is in epidermal electronics, e.g., self-adhering bioelectronic decal monitoring of the concentration of cations (Na^+^, K^+^, and Ca^2+^) in sweat as a marker of the physiological status of the organism [46]. 

The use of bacterial cellulose in skin tissue engineering and wound healing, including its clinical applications, has been reviewed by Fu et al. [45].

### 5.2. Plant- and Animal-Derived Nanocellulose in Wound Healing

#### 5.2.1. Plant-Derived Nanocellulose without Additives

Like bacterial nanocelullose, plant cellulose can also appear in the form of a hydrogel containing nanofibrils, and can induce wound healing by itself, i.e., without any additives. Wood-derived nanofibrillar cellulose (NFC) has been tested for wound dressing applications due to its high capability to absorb liquids and to form translucent films. These properties are required for non-healing and chronic wounds, where exudates need to be managed adequately. In addition, the translucency of NFC makes it possible to evaluate the development of the wound without needing to remove the dressing [143]. The healing potential of wood-derived NFC was tested in a clinical trial on burn patients. An NFC dressing was applied to split thickness skin graft donor sites. The NFC dressing was compared with the Suprathel® commercial lactocapromer dressing (PMI Polymedics, Denkendorf, Germany). Epithelialization of the donor site was faster when covered by the NFC dressing than when Suprathel® was used. The NFC dressing seemed to be promising for skin graft donor site treatment, since it was biocompatible, it attached easily to the wound bed, and it remained in place until the donor site had renewed. It also detaches from the epithelialized skin by itself [60]. 

Wood-derived NFC (obtained from commercial never-dried bleached sulfite softwood dissolving pulp), crosslinked with calcium ions, also had hemostatic potential, especially when enriched with kaolin or collagen [63]. In addition, inflammatory response studies with blood-derived mononuclear cells revealed the inert nature of NFC hydrogels in terms of cytokine secretion and reactive oxygen species production. Water retention tests showed the potential of NFC hydrogels to maintain a suitably moist environment for various types of wounds [26].

Hemostatic potential was also observed in oxidized cellulose (OC) modified in an inert argon plasma [274]. The plasma-modified OC was more acidic, and had a larger surface area and greater ability to absorb water. These factors are crucial for effective haemostasis. In addition, the acidity of plasma-modified OC increased its antibacterial activity. Plasma-modification could therefore be utilized for advantageous modification and also for sterilizing the OC haemostat in a single easy step [274].

Similarly, nanofibrillated cellulose, prepared from *Pinus radiata* pulp fibers and pre-treated with TEMPO-mediated oxidation, in the form of films, impaired the growth of *Pseudomonas aeruginosa*, a frequent wound pathogen, and led to more death of bacterial cells than Aquacel®, a commercial control wound dressing [65]. The same NFC in suspension and in the form of aerogels also showed activity against *Pseudomonas aeruginosa* PAO1. In the case of aerogels, bacterial biofilm formation decreased with decreasing porosity and surface roughness of the material [275]. Incorporating cellulose nanocrystals (derived from *Hibiscus cannabinus*) into chitosan/poly(vinyl pyrrolidone) composite membranes, developed for wound dressing applications, enhanced their antibacterial activity, as revealed in *Staphylococcus aureus* and *Pseudomonas aeruginosa* [72]. The antibacterial activity was further increased by coating these membranes with hydrophobic stearic acid, which hampered the adhesion of bacterial cells [276].

#### 5.2.2. Plant-Derived Nanocellulose with Additives

As in bacterial nanocellulose, the antibacterial properties of plant-derived nanocellulose can be further improved by various chemical modifications and by adding various ions, nanoparticles, and synthetic or nature-derived molecules. An example of chemical modification is carboxymethylation and periodate oxidation of nanocellulose, which was then used as bioink for preparing porous antibacterial wound dressings [277]. As concerns the ions, the antibacterial properties of softwood pulp–derived NFC were modulated by using divalent calcium or copper ions as crosslinking agents. Calcium-crosslinked hydrogels were more active against *Pseudomonas aeruginosa*, while copper-crosslinked hydrogels were more active against *Staphylococcus epidermidis* [27]. In addition, Ca^2+^-crosslinked NFC hydrogels could be used for topical drug delivery applications in a chronic wound healing context [64]. Copper-containing nanocellulose materials also showed angiogenic activity. Composites of wood-derived NFC and copper-containing mesoporous bioactive glass showed not only antibacterial activity against *Escherichia coli*, but also angiogenic activity, as revealed in a 3D spheroid culture system of human umbilical vein endothelial cells, and also in cultures of mouse 3T3 fibroblasts, which upregulated the expression of pro-angiogenic genes in these cells [278]. Nanocomposite hydrogels containing carboxylated cellulose nanofibers (prepared by TEMPO-mediated oxidation), gelatin and aminated silver nanoparticles showed strong mechanical and self-recovery properties, antibacterial activity against *Staphylococcus aureus* and *Pseudomonas aeruginosa*, satisfactory hemostatic performance, and an appropriate balance of fluids on the bed of skin wounds created in mice [6]. Micro- and nanofibrillated cellulose incorporated with bismuth complexes was effective against *Escherichia coli*, *Staphylococcus aureus*, methicillin-resistant *Staphylococcus aureus*, and vancomycin-resistant *Enterococcus* [279]. 

Examples of synthetic and nature-derived molecules that have been incorporated into plant-derived nanocellulose in order to enhance its antibacterial activity include antibiotics, antiseptics, antimicrobial peptides, alkanin, shikonin, isoliquiritigenin, coumarin, and curcumin. From this point of view, nanocellulose-based materials can serve as carriers for topical and transdermal drug delivery. For example, a gentamycin-grafted nanocellulose sponge, prepared by multi-crosslinking of CNF (extracted from wood pulp), cellulose acetoacetate and 3-aminopropyl(triethoxy)silane, showed excellent antibacterial performance against *Escherichia coli* and *Staphylococcus aureus*, with bactericidal rates of over 99.9% [41]. Similarly, a hydrogel containing cellulose nanofibrils (produced by TEMPO-mediated oxidation from bleached softwood kraft pulp), and polydopamine loaded with tetracycline, was active against *Escherichia coli* and *Staphylococcus aureus*, and stimulated the healing of skin defects created in rats in vivo [6]. Due to the increasing resistance to antibiotics, attention has also been paid to other antimicrobial compounds. Composite films containing spherical cellulose nanocrystals and titania nanoparticles complexed with triclosan, i.e., an antibacterial and antifungal agent, showed activity against *Escherichia coli* and *Staphylococcus aureus* [280]. Another novel strategy for fighting bacterial infections involves delivering antibacterial peptides, e.g., nisin, a polycyclic antibacterial peptide produced by the bacterium *Lactococcus lactis*. This peptide was incorporated into TEMPO-oxidized nanofibrillated cellulose (TONFC) via electrostatic attraction between the negatively-charged TONFC surface and the positively-charged nisin molecules. The capacity of TONFC to bind nisin was regulated by pH and ionic strength. The activity against *Bacilus subtilis* and *Staphylococcus aureus* was higher in nisin-TONFC composites than in free nisin [135]. In another nanocellulose-based material, i.e., nanocrystalline cellulose functionalized with aldehyde groups, also known as sterically stabilized nanocrystalline cellulose, nisin was combined with lysozyme, another antibacterial agent [102]. Other interesting molecules are alkannin, shikonin, and their derivatives, which are naturally occurring hydroxynaphthoquinones with wound healing potential and antimicrobial, anti-inflammatory, antioxidant, and antitumor activities. In a study by Kontogiannopoulos et al. [136], these agents were for the first time incorporated in electrospun cellulose acetate nanofibrous meshes for potential wound dressings. Isoliquiritigenin, a phenolic compound found in licorice, was incorporated into pH-sensitive hydroxyethyl cellulose/hyaluronic acid complex hydrogels. These composites showed antimicrobial activity against *Propionibacterium acnes*, and they were therefore considered to be promising for treating acne [281]. 

Other important plant-derived molecules for incorporation into wound dressings are coumarin and curcumin. These compounds have a wide spectrum of biological and pharmacological activities, including antioxidant, anti-inflammatory, antimicrobial and anticancer activities. However, as was mentioned above, their potential therapeutic applications are hindered by the low stability and the poor water-solubility of these molecules. Attempts have been made to overcome these drawbacks and to improve the bioavailability of these compounds, e.g., using a Pickering emulsion, i.e., a kind of emulsion stabilized by solid particles located at the oil-water interface, in which aminated nanocellulose particles were used [137]. Another approach was a nanocellulose-reinforced chitosan hydrogel incorporated with Tween 20, i.e., a non-ionic surfactant, in order to improve the solubility of curcumin [282]. Other relatively simple cellulosic materials for curcumin delivery include capsules made of ethyl cellulose blended with methyl cellulose [283]. More complicated materials are polyvinyl alcohol/polyethylene oxide/CMC cellulose matrix blended with nanosilver nanohydrogels, Aloe vera and curcumin, deposited on a hydrolysed PET fabric [284], electrospun nanofibers containing PLGA, cellulose nanocrystals, curcumin and polyethyleneimine-carboxymethyl chitosan/pDNA-angiogenin nanoparticles [285], and composites made of complexes of curcumin/gelatin microspheres and porous collagen-cellulose nanocrystals [286].

Another important material with antimicrobial activity is chitosan. Chitosan and pectin with organic rectorite, i.e., a layered silicate, were used for deposition on electrospun cellulose acetate nanofibers in order to inhibit bacterial growth, which was proven on *Escherichia coli* and *Staphylococcus aureus.* At the same time, the material supported the growth of human epidermal cells. This material was considered to be suitable for wound dressing and food packaging [28]. In a study by Vosmanská et al. [287], a three-step modification of the standard cellulose wound dressing was prepared. This modification included argon plasma-treatment, chitosan impregnation and AgCl precipitation. The plasma treatment oxidized the material surface, which increased the hydrophilicity of the material surface and the amount of chitosan impregnated on to the surface. In addition, plasma treatment almost doubled the amount of AgCl precipitated on the plasma-activated surface. All these factors endowed the cellulose-based wound dressing with antibacterial activity against *E. coli* and *S. epidermidis* [287].

Various antibacterial nanocellulose-based materials have been reviewed by Li et al. [144].

Nanocellulose in the form of nanocrystals has been widely used for delivering drugs for wound healing and for treating various skin disorders. Cellulose nanocrystals conjugated with folic acid are promising vectors for the targeted delivery of chemotherapeutic agents to folate receptor-positive cancer cells [288]. Cellulose nanocrystals (CNCs) isolated from *Syzygium cumini* leaves or bamboo, impregnated with silver nanoparticles, have been proposed for accelerated healing of acute and diabetic wounds [68,77]. Other potential wound dressings for accelerated healing of diabetic wounds are composite nanofibrous membranes containing PLGA fibres and cellulose nanocrystals loaded with neurotensin, an inflammatory modulator [289]. Cellulose nanocrystals loaded with hydroquinone, which inhibits the production of melanin, were designed for treating hyperpigmentation, a disorder occurring during pregnancy and sun exposure [290]. The use of various cellulose-based nanocarriers, such as bacterial cellulose, cellulose acetate, microcrystalline cellulose, CMC, cellulose nanocrystals, cellulose nanofibrils, etc., in drug delivery systems for cancer treatment has been reviewed in [291]. Advanced “intelligent” nanocellulose-based wound dressings were combined with biosensors, e.g., for human neutrophil elastase present in chronic wound fluid [121,292,293].

#### 5.2.3. Animal-Derived Nanocellulose

Animal-derived nanocellulose also has potential for application in wound dressings. Cellulose membranes manufactured from *Styela clava* tunics, by themselves and in combination with alginate or selenium, stimulated healing of surgically created wounds in normal rats and in rats with diabetes induced by treatment with streptozotocin [81,90], probably through regulation of angiogenesis and connective tissue formation.

## 6. Potential Cytotoxicity and Immunogenicity of Nanocellulose

Nanocellulose materials are often considered as materials with no cytotoxicity and immunogenicity, or with low cytotoxicity and immunogenicity. Cellulose nanofibers isolated from Curauá leaf fibers (*Ananas erectifolius*) provide an example of non-cytotoxicity. They showed no signs of cytotoxicity in direct or indirect assays for cell viability and cell morphology using Vero cells, i.e., monkey-derived kidney epithelial cells. Moreover, during an adhesion test, the cells demonstrated a relatively high affinity to the CNF surface [15]. Cotton-derived cellulose nanocrystals (mean width 7.3 nm, mean length 135 nm, concentrations from 30 to 300 µg/µl per ml of cell culture medium) are an example of non-immunogenic nanocellulose. These nanocrystals did not induce any release of pro-inflammatory cytokines, namely tumor necrosis factor-α (TNF-α) and interleukin-1β (IL-1β), from human macrophages derived from peripheral blood monocytes, while microcrystalline cellulose (particle size ~50 µm) induced the release of these cytokines [32]. 

However, several studies documenting considerable cytotoxicity and pro-inflammatory activity of nanocellulose in vitro and in vivo have also emerged. In vitro, five types of wood-derived nanocellulose materials (doses up to 100 µg/ml of cell culture medium) were practically non-cytotoxic for human macrophage-like THP-1 cells, when compared with multi-walled carbon nanotubes and nanomaterials based on ZnO, Ag and SiO_2_, as revealed by an Alamar blue assay. However, multiplex profiling of cytokine and chemokine secretion indicated that nanocellulose materials induced potent inflammatory responses at sub-cytotoxic doses [294]. In vivo, wood-derived cellulose nanocrystals were shown to induce an inflammatory response in mice after aspiration, manifested by an increase in leukocytes and eosinophils in the lungs, recovered by bronchoalveolar lavage (BAL), and up-regulation of pro-inflammatory cytokines and chemokines, such as TNF-a, G-CSF, GM-CSF, INF-γ, MCP-1, MIP-1α, MIP-1β, RANTES, and various interleukins (including IL-1β), in the BAL fluid. These nanocrystals also induced oxidative stress and tissue damage, manifested by an accumulation of oxidatively modified proteins and an increase in lactate dehydrogenase activity in BAL fluid [17]. Similar results were obtained in a study by Shvedova et al. [4]. The exposure of mice to respirable wood-derived cellulose nanocrystals caused pulmonary inflammation and damage, induced oxidative stress, increased levels of collagen and transforming growth factor- β (TGF-β) in the lung, and impaired pulmonary functions. In addition, these effects were more pronounced in female mice than in male mice [4]. Sulphonated nanocellulose obtained from *Khaya sengalensis* seed showed renal toxicity in rats, manifested by hypernatremia, enhancement of the antioxidant status and immunohistochemical expressions of inducible nitric oxide synthase (iNOS) and cyclooxygenase-2 (COX-2) in the kidneys [70].

The cytotoxicity and immunogenicity of nanocellulose can be modulated by its physicochemical properties, e.g., by functionalizing it with specific chemical groups or by endowing it with an electrical charge. Wood-derived nanofibrillated cellulose (NFC) modified with carboxymethyl groups (anionic nanocellulose) and hydroxypropyltrimethylammonium groups (cationic nanocellulose) elicited a lower pro-inflammatory effect than unmodified NFC in human dermal fibroblasts, in lung MRC-5 cells and in human macrophage-like THP-1 cells [148]. However, the anionic NFC films significantly activated THP-1 cells toward a pro-inflammatory phenotype, whereas the cationic and unmodified cellulose induced only mild activation of these cells [150].

The morphology of cellulose nanoparticles can also influence their cytotoxicity and immunogenicity. Nanofibrillated cellulose (NCF) showed more pronounced cytotoxicity and oxidative stress responses in human lung epithelial A549 cells than cellulose nanocrystals (CNC). However, exposure to CNC caused an inflammatory response with significantly elevated pro-inflammatory cytokines and chemokines compared to NCF. Interestingly, cellulose staining indicated that CNC particles, but not NCF particles, were taken up by the cells [295]. In vivo experiments performed in mice also confirmed different immune responses to NFC and to CNC. Pulmonary exposure to NFC led to discrete local immune cell polarization patterns with TH1-like immune reaction, while CNC caused non-classical or non-uniform responses. However, the response to both types of nanocellulose was milder than the response to asbestos and carbon nanotubes [296]. In addition, curcumin was able to suppress, at least in part, the immune response to cationic needle-like cellulose nanocrystals, as observed by diminished IL-1β secretion in mouse J774A.1 macrophages primed with LPS [18]. The immunogenicity of bacterial, wood-based, and algal nanocellulose may also be because these types of nanocellulose can contain immunogenic contaminants, such as endotoxin and (1,3)-β-d-glucan [85,297]. 

## 7. Conclusions

Nanocellulose is a promising material for a wide range of applications in industry, technology, biotechnology, and medicine, including tissue engineering and wound healing. However, the non-degradability of nanocellulose in the human organism is a limiting factor for its direct use in skin tissue engineering as a scaffold for skin cells, because scaffolds persisting in the skin could lead to scar formation and other complications. A more promising approach is therefore to use nanocellulose as a temporary carrier for delivering cells into wounds, which can be removed after the cells have adhered to the wound bed. However, artificial skin constructed in vitro could be used for experimental purposes, e.g., for studies on the biology, metabolism, and vascularization of skin tissue, and for studies on the effects of various drugs, as was demonstrated in artificial liver, adipose, and tumor tissues. In skin applications, nanocellulose seems to hold the greatest promise as an advanced dressing material for topical, transdermal, and systemic applications of various drugs, as a transparent dressing material enabling direct inspection of wounds, as a dressing material coupled with sensors, and as a material for constructing epidermal electronics. 

## Figures and Tables

**Figure 1 nanomaterials-09-00164-f001:**
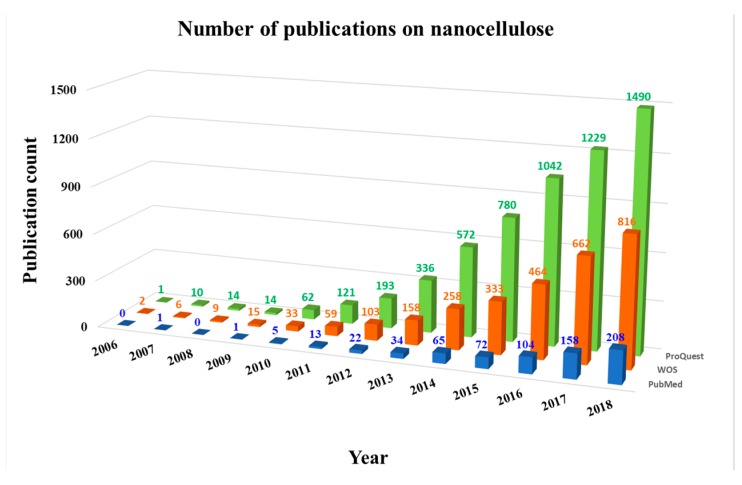
The number of publications on nanocellulose found in three databases: ProQuest (green), Web of Science (WOS, red), and PubMed (blue) from 2006 to 2018 using the search term “nanocellulose.”

**Figure 2 nanomaterials-09-00164-f002:**
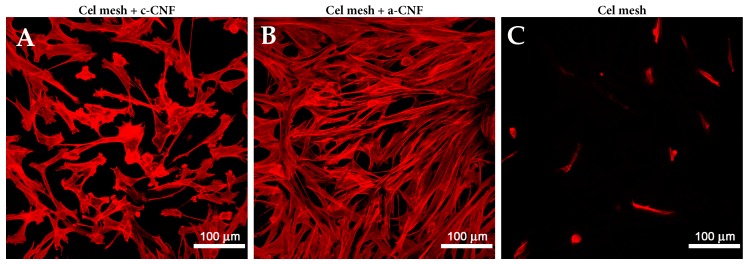
Morphology of human dermal fibroblasts on day 4 after seeding on a cellulose mesh modified with cationic cellulose nanofibers (**A**), with anionic cellulose nanofibers (**B**), and on pristine cellulose mesh (**C**). The cells were stained with phalloidine conjugated with TRITC (stains F-actin, red fluorescence). Leica TCS SPE DM2500 confocal microscope (Leica Microsystems, Wetzlar, Germany), obj. 20×/1.15 NA oil.

**Figure 3 nanomaterials-09-00164-f003:**
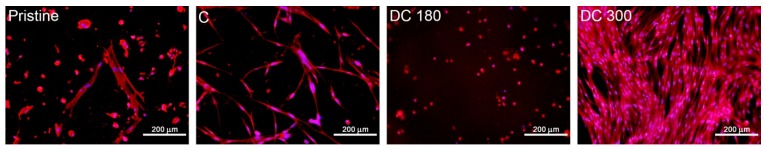
Human dermal fibroblasts on bacterial cellulose in a pristine state (Pristine), loaded with curcumin (C) or with degraded curcumin at 180 °C (DC 180) or at 300°C (DC 300) after seven days of cell seeding. The cells were stained with Texas Red C2-Maleimide (red fluorescence, cell membrane and cytoplasm) and Hoechst #33258 (blue fluorescence, cell nuclei). Olympus IX 50 microscope, obj. 10x, DP 70 digital camera (Olympus, Tokyo, Japan).

**Table 1 nanomaterials-09-00164-t001:** Types of nanocellulose.

Nanocellulose structures	Example	Dimensions	Reference
Nanofibrils	In bacterial cellulose	Diameter from 70 to 140 nm, length in µm	[2]
	In wood-derived cellulose	Diameter 3–5 nm, length several µm, form 20–50 nm thick aggregates	[26,27]
Nanofibers	Created by electrospinning	Cellulose acetate: average diameterabout 400 nm	[28]
		Bacterial cellulose (33 wt. %) with chitosan: diameters from 80 to 170 nm	[29]
	Isolated from pineapple	Width 6.4 ± 4.6 nm, length in µm	[15]
Nanowhiskers	Pine kraft pulp	Diameter 5–15 nm, length 100–250 nm	[24]
	Kenaf bast	Diameter 10–15 nm, length hundreds nm	[30]
	Bacterial cellulose	Diameter 10–100 nm, length 100–1000 nm	[31]
Nanocrystals	Cotton-derived	Mean width 7.3 nm, mean length 135 nm	[32]
Nanorods	Grass-derived	Width 15 ± 3 nm, length 120 ± 15 nm	[19]
Nanoballs	Wood-derived	Diameter 80-85 nm	[23,24]
Nanoplatelets	Agave-derived	Thickness 80 nm, other dimensions in µm	[25]

**Table 2 nanomaterials-09-00164-t002:** Industrial and (bio)technological applications of nanocellulose.

Application	Specification	Example	Reference
Adsorption	Air purification	Odor removal (in combination with zeolites)	[94]
	Removal of pollutants from aqueous solutions	Heavy metal ions (Cu^2+^, Pb^2+^, Hg^2+^)	[88,95]
		Toxic dyes (methylene blue, Congo Red)	[38,96]
		Mefenamic acid (a nonsteroidal anti-inflammatory drug, a potential endocrine disruptor)	[97]
		Oily substances	[39,93]
		Insecticides (neonicotinoids in milk)	[98]
	Immobilization of atoms and (bio) molecules	Metal catalysts (copper)	[99]
		Proteins (bovine serum albumin, lysozyme, γ-globulin, and human IgG	[86]
		Enzymes (trypsin, laccase, lysozyme, lipase)	[69,100,101,102]
		Ingested lipids (obesity management)	[103]
		DNA oligomers	[104]
(Ultra)filtration	Removal of toxic dyes	Methylene blue, methylene orange, rhodamine	[105]
	Hemodialysis membranes	Nanofibrillated cellulose with polypyrrole	[106]
	Removal of viruses	Swine influenza virus	[83]
		Murine leukemia virus	[107]
		Bacteriophages	[87]
Packaging	Food, sensitive devices	Self-standing nanocellulose films from birch pulp	[33]
		Paper sheets modified with nanocellulose and chitosan	[108]
Conservation	Historical papers, cotton canvas	Cellulose nanofibrils, carboxymethylated cellulose nanofibrils, cellulose nanocrystals	[109]
Thermal applications	Thermal insulators	Wood-derived nanofibrils with extremely low thermal conductivity	[110]
	Fire retardants	Wood-derived cellulose nanofibrils with silica nanoparticles	[111]
		Wood-derived nanocellulose with montmorillonite clay	[67]
Energy extraction and storage	Lithium batteries	Nanocellulose/polypyrrole	[112]
		Nanocellulose/polyethylene	[113]
		Graphene/nanocellulose/silicon	[114]
	Solar cells/panels	Nanofibers from sisal with graphene oxide	[81]
	(Super)capacitors	Bacterial nanocellulose/carbon nanotubes/triblock-copolymer ion gels	[115]
		Nanocellulose with polyaniline	[116]
Acoustics	Membranes for loudspeakers	Cellulose nanofibers with Fe_3_O_4_ nanoparticles	[79]
(Bio)sensors	Optical SERS-based	Detection of pesticides, dyes, bacteria	[117,118]
	Optical fluorescence-based	Detection of heavy metals	[119]
		Detection of thiols	[120]
		Detection of elastase	[121]
	Chemical	Detection of vapors (NH_3_.H_2_O, H_2_O, HCl, acetic acid)	[19]
	Electrochemical	Detection of cations in biological fluids (Na^+^, K^+^, Ca^2+^)	[46]
		Detection of cholesterol	[122]
		Detection of avian leukosis virus	[123]
	Piezoelectric	Based on bacterial cellulose	[124]
		Based on plant-derived cellulose nanofibrils	[125]
		Based on nanocellulose with chitosan	[126]
		Tactile sensor (simultaneous sensing of temperature and pressure)	[127]
		Strain-sensing protonated aerogels from cellulose nanofibrils	[128]
Drug delivery	Peroral	Paracetamol	[57]
		Ibuprofen (colonic release)	[78]
		Methotrexate (colonic release)	[129]
	Transdermal	Analgesics, antiphlogistics, corticoids, antihypertensives	[58,59]
		Diclophenac	[130]
		Propranolol	[131]
	Topical	Local anesthetics	[132]
		Antiseptics	[133,134]
		Antibiotics (gentamycin, ceftriaxone)	[41,51]
		Antibacterial peptides	[135]
		Other antimicrobial, anti-inflammatory and antitumor drugs	[136,137]

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
