# Peer review of "Versatile Application of Nanocellulose: From Industry to Skin Tissue Engineering and Wound Healing"

_nanomaterials, 2019, doi:10.3390/nano9020164_

Round 1

Reviewer 1 Report

The review focused on nanotechnology, the authors have reflected a good work about the evolution of this technology to finally writte  about the nanocellulose in skin tissue

However, I do have some minor comments that would be useful to improve the clarity of the work:

At the abstract, the authors use “e.g.” three times. Maybe they can use others alternatives: such as, for example….

About the point 3, could be summarized, since the purpose of the revision is centered on application of nanocellulose in skin tissue.

Author Response

Comments and Suggestions for Authors

The review focused on nanotechnology, the authors have reflected a good work about the evolution of this technology to finally writte  about the nanocellulose in skin tissue

However, I do have some minor comments that would be useful to improve the clarity of the work:

At the abstract, the authors use “e.g.” three times. Maybe they can use others alternatives: such as, for example….

Reply: Other alternatives have been used in the abstract instead of “e.g.”, namely the following formulations:

“nanofibrils, found in bacterial cellulose or wood”

“nanofibers, present particularly in electrospun matrices”

“…..a wide range of industrial, technology and biomedical applications, namely for adsorption, ultrafiltration………”

About the point 3, could be summarized, since the purpose of the revision is centered on application of nanocellulose in skin tissue.

Reply: In fact, the point 3 “Recent use of nanocellulose in tissue engineering and tissue repair” is summarized, because it contains only relatively brief paragraphs to each type of tissue engineering (neural tissue engineering, cartilage tissue engineering, liver tissue engineering…….). In contrast, the skin tissue engineering and wound healing, on which the review is mainly focused, is treated much more deeply in two long sections subdivided into  4-7 subsections, each containing several paragraphs. The authors of the article wished to present the applications of nanocellulose in skin issue in context with application of this material in other tissues, and to show how broad can be the use of nanocellulose in tissue engineering, tissue repair and other related fields.

Reviewer 2 Report

In this manuscript, Bacakova et al. presented an overview of different types of nanocellulose and nanocellulose based materials.

Authors have provided an introduction to different types of nanocelluloses, various sources and their application in various fields.  Further, authors have focused on various reports on the application of nanocellulose in tissue engineering and biomedical field.

Nanocellulose and nanocellulose based materials are evolving as an important source of natural and renewable materials for potential application in technology, sensors, water purification, tissue engineering, wound healing and photonics.  Therefore, the topic presented in this manuscript of interest for a wider range of readers.  However, recently there have been several reviews on nanocelluloses with more clear and comprehensive description.  Therefore, I am wondering whether the present review brings anything new to this field?

Comments/Corrections:

The title, abstract, introduction do not really in agreement with each other.  In the introduction, authors have focused mostly on types of nanocellulose, its materials properties, and application in various fields other than tissue engineering.

There is no explanation on either in the abstract nor in the introductions on why there is need for skin tissue engineering and how nanocellulose is going to solve that problem or what properties of nanocellulose will do that?

Nanocelluloses are prepared with various functional groups/surface modification that can alter the physicochemical properties dramatically.  No discussion was found on this topic

Figure 2 in the caption, It should be "Various sources of nanocellulose" not "various examples"

In page 8, the history of nanocellulose is completely misleading.  PubMed is not the best database for search.  I recommend authors to use SciFinder or Web of Science.  The first paper appeared is not 2007, instead in 2006 (see:Nanocellulose polymer composites as innovative pool for (bio)material development By: Kramer, Friederike; Klemm, Dieter;Schumann, Dieter; et al. Conference: Conference on New Cellulose Products and Composites Location: Wiesbaden, Germany Date: JUN 26-29, 2006 MACROMOLECULAR SYMPOSIA   Volume: 244   Pages: 136-148   Published: 2006)

The graph showing the number of publications is completely wrong as the recent publications exceed 500+ (see: 2018 (785)  2017 (649)  2016 (442)  2015 (313).

The present version looks more like a report, rather than a review.  Authors should focus on selected topics in tissue engineering with proper figures and better discussion on each topic.

Important reference should be includedKonturi et al. Advanced Materials through Assembly of Nanocelluloses. Adv. Mater. 2018, 30, 1703779).

Author Response

Comments and Suggestions for Authors

In this manuscript, Bacakova et al. presented an overview of different types of nanocellulose and nanocellulose based materials.

Authors have provided an introduction to different types of nanocelluloses, various sources and their application in various fields.  Further, authors have focused on various reports on the application of nanocellulose in tissue engineering and biomedical field.

Nanocellulose and nanocellulose based materials are evolving as an important source of natural and renewable materials for potential application in technology, sensors, water purification, tissue engineering, wound healing and photonics.  Therefore, the topic presented in this manuscript of interest for a wider range of readers.  However, recently there have been several reviews on nanocelluloses with more clear and comprehensive description.  Therefore, I am wondering whether the present review brings anything new to this field?

Reply: Recently, several reviews on the application of nanocellulose in tissue engineering and related biotechnologies, particularly drug delivery, have emerged, and most of them are cited in our article. However, to the best of our knowledge, none of them is specialized or at least deeply focused on skin tissue engineering and wound healing. Our review is not fully specialized to these topics, too, but it summarizes almost all recent knowledge on the use of nanocellulose in these fields in context with other fields of tissue engineering, other biotechnologies, such as drug delivery, and also in context with other technological and industrial applications of nanocellulose.  We believe that a certain breadth of our review could attract a wide community of readers, not only biologists and tissue engineers, and it could considerably help scientists to find their way in the broad field of nanocellulose research. Last but not least, we anticipate that the broad spectrum of aspects included in our review would increase the number of citations of this review.

Comments/Corrections:

The title, abstract, introduction do not really in agreement with each other.  In the introduction, authors have focused mostly on types of nanocellulose, its materials properties, and application in various fields other than tissue engineering.

Reply: In the Introduction section, the authors of this review considered that it is important to define the nanocellulose and to mention its types, sources, properties and applications in general, i.e. not only in tissue engineering, but also in various fields of industry, technology and biotechnology other than tissue engineering. These applications of nanocellulose are often better known in the scientific community than the applications in tissue engineering, and therefore the authors wished to introduce the use of nanocellulose in tissue engineering in a broader context of other interesting applications. Nevertheless, the referee is right that little is mentioned in the Introduction on the use of nanocellulose in tissue engineering. Therefore, the use of nanocellulose in various fields of tissue engineering was introduced more deeply, including explanation of the reasons of this use, which lie in several advantageous properties of nanocellulose. In addition, in order to achieve a higher agreement among the title, abstract and introduction, a mention on non-biomedical application of nanocellulose was added in the title of the article. The title of the article was changed from “Nanocellulose in biotechnology and medicine: focus on skin tissue engineering and wound healing” to “Versatile application of nanocellulose: from industry to skin tissue engineering and wound healing”.

There is no explanation on either in the abstract nor in the introductions on why there is need for skin tissue engineering and how nanocellulose is going to solve that problem or what properties of nanocellulose will do that?

Reply: The skin is the largest organ of the human body with several vitally important functions, particularly as barrier against adverse effects of the surrounding environment on the organism (chemical damage, radiation damage, e.g. by UV light, microbial infection). Other important functions of skin include thermoregulation, sensation of temperature, touch, pressure and pain, keeping appropriate moisture in the underlying tissues, excretion of ions, water and various molecules (e.g. lipids and  proteins), and also production and storage of various biomolecules, such as pigments, vitamin D and keratins for formation of epidermal appendages. Therefore, there is essential need to regenerate or at least to repair the damaged skin, particularly by methods of skin tissue engineering and induction of active wound healing. Nanocellulose has several advantageous properties for these applications, such as appropriate mechanical strength, high water-absorbing capacity, which enables to keep the moisture in the damaged skin, and at the same time, to absorb the exudate from the wounds. The nanoscale morphology of the nanocellulose imitates the nanoarchitecture of the native extracellular matrix, and thus the nanocellulose can be regarded as a suitable substrate for the adhesion and growth of skin cells. Also other physicochemical properties of nanocellulose, such as its wettability and electrical charge, can be tailored by functionalization with various chemical groups or by preparation of nanocellulose from various sources and by various methods. Last but not least, some types of nanocellulose, e.g. wood-derived nanocellulose, have antimicrobial effect, or this effect can be induced by incorporation of nanocellulose materials with various ions and compounds. On the other hand, the use on nanocellulose in skin tissue engineering is limited by its non-degradability in the human organism. Persistence of nanocellulose materials in skin tissue constructs can lead to scar formation. Therefore, as recommended in this review, it would be better to use the nanocellulose materials as temporary carriers for skin cells, removable after the cell delivery into the damaged skin, and not as direct scaffolds for cells in tissue-engineered skin constructs.

These aspects were included in the Introduction section of the manuscript, and are also discussed in the following sections of the manuscript.

Nanocelluloses are prepared with various functional groups/surface modification that can alter the physicochemical properties dramatically.  No discussion was found on this topic

Reply: The paragraph on the modification of nanocellulose with various functional groups and its impact on the physicochemical properties of nanocellulose, particularly on its wettability and electrical charge, was added into the Introduction. These modifications, which have a great impact on the cell behavior, such as the cell adhesion, proliferation, differentiation, viability, morphology, orientation and immune activation, are then discussed (and most of them were originally discussed) in other parts of the article, namely in the section 3 (paragraphs on neural tissue engineering, bone implant coating and cell transfection), the subsection 4.2. (the first paragraph and other paragraphs dealing with our own results), the subsections 5.1.2. and 5.2.2. (sentences dealing with the hydrophilicity of nanocellulose), the section 6 (paragraph 3).

Figure 2 in the caption, It should be "Various sources of nanocellulose" not "various examples"

Reply: The Figure 2 with its caption was deleted due to copyright problems.

In page 8, the history of nanocellulose is completely misleading.  PubMed is not the best database for search.  I recommend authors to use SciFinder or Web of Science.  The first paper appeared is not 2007, instead in 2006 (see:Nanocellulose polymer composites as innovative pool for (bio)material development By: Kramer, Friederike; Klemm, Dieter;Schumann, Dieter; et al. Conference: Conference on New Cellulose Products and Composites Location: Wiesbaden, Germany Date: JUN 26-29, 2006 MACROMOLECULAR SYMPOSIA   Volume: 244   Pages: 136-148   Published: 2006)

Reply: The PubMed database was originally selected as a database widely known and used by biologists, physicians and other scientists dealing with various biotechnologies. However, the referee is right that this database is relatively narrow, limited to articles in journals with impact factor, and usually not containing papers in conference proceedings, book chapters on other types of publications. In addition, some journals, focused on physics, chemistry, material engineering, industry and technology are missing in this database. Therefore, the results from searching in other databases, particularly the Web of Science (WOS), were included to the section “History of nanocellulose research” and to the graph. The database “SciFinder” was not accessible for us, thus we used the results from another large database, i.e. ProQuest. The paper by Kramer et al. (2006) was added to the manuscript.

The graph showing the number of publications is completely wrong as the recent publications exceed 500+ (see: 2018 (785)  2017 (649)  2016 (442)  2015 (313).

Reply: The graph is not wrong, only it is limited to the PubMed database, from which the numbers of publications for construction of the graph were extracted. The numbers of publications provided by two other databases, i.e. WOS and ProQuest, were added to the graph.

The present version looks more like a report, rather than a review.  Authors should focus on selected topics in tissue engineering with proper figures and better discussion on each topic.

Reply: The authors tried to focus on tissue engineering, particularly skin tissue engineering, as much as possible, but at the same time, they wanted to show this topic in context with other applications of nanocellulose, and also with the types, sources, properties and other characteristics of this (fascinating) material. The Figure 1, demonstrating the morphology of various types of nanocellulose, and Figure 2, showing various sources of nanocellulose, were deleted due to copyright problems, and only figures related to skin tissue engineering and wound healing of our own provenience were left in the manuscript. The reasons for the use of nanocellulose in skin tissue engineering and wound healing were better explained in the Introduction, and the discussion of these topics was improved in the following parts of the manuscript.

Important reference should be included Konturi et al. Advanced Materials through Assembly of Nanocelluloses. Adv. Mater. 2018, 30, 1703779).

Reply: This reference was added to the Introduction section – into paragraphs dealing with the types of nanocellulose and its physicochemical properties.

Reviewer 3 Report

This work presents an exhaustive review on the applications of nanocellulose in biotechnology and medicine, with a main focus on skin tissue engineering and wound healing. The topic is interesting and the article is well structured and well written. I recommend its publication in Nanomaterials after addressing the following points:

1) PubMed does not seem to provide all the relevant literature on nanocellulose. For example, the following publications are missing:

- Ahola et al. Biomacromolecules 9 (2008), 1273-1282 (one of the initial works on nanocellulose)

- Bhattacharya et al. Journal of Controlled Release 164 (2012), 291–298

- Lou et al. Stem Cells and Development 23 (2014), 380-392

- Orelma et al. Biomacromolecules 13 (2012), 2802-2810

And so on… I understand that it would be impossible to cite all the publications related to nanocellulose, but it seems that the numbers provided in Figure 3 underestimate the reality.

2) Although the terminology on nanocellulose materials could be a bit confusing, it is often considered that cellulose whiskers, cellulose nanocrystals and cellulose nanorods are synonyms (See the review by Klemm et al. Angew. Chem. Int. Ed. 50 (2011), 5438-5466).

3) The scale bars in Figure 1 A-C are not visible.

4) Lines 569-570 read: “BNC itself, i.e. without any additives, showed a great capacity to stimulate wound healing, i.e. regeneration of the epidermis and dermis.” But lines 720-721 read: “Unlike bacterial nanocelullose, however, plant cellulose can induce wound healing by itself, i.e. without any additives.” These sentences contradict each other regarding the capacity of bacterial nanocellulose itself to induce wound healing.

5) Some typos to be corrected are:

- Line 21: Gluconacetobacter

- Line 225: SH-SY5Y

Author Response

Comments and Suggestions for Authors

This work presents an exhaustive review on the applications of nanocellulose in biotechnology and medicine, with a main focus on skin tissue engineering and wound healing. The topic is interesting and the article is well structured and well written. I recommend its publication in Nanomaterials after addressing the following points:

1) PubMed does not seem to provide all the relevant literature on nanocellulose. For example, the following publications are missing:

- Ahola et al. Biomacromolecules 9 (2008), 1273-1282 (one of the initial works on nanocellulose)

- Bhattacharya et al. Journal of Controlled Release 164 (2012), 291–298

- Lou et al. Stem Cells and Development 23 (2014), 380-392

- Orelma et al. Biomacromolecules 13 (2012), 2802-2810

And so on… I understand that it would be impossible to cite all the publications related to nanocellulose, but it seems that the numbers provided in Figure 3 underestimate the reality.

Reply: The review article was enriched with data from other databases, namely Web of Science and ProQuest, which provided much more matches after placing the term “nanocellulose”. The requested publications (and many others) were included into the article with appropriate text (please, see also the reply to the comments by Reviewer 2).

2) Although the terminology on nanocellulose materials could be a bit confusing, it is often considered that cellulose whiskers, cellulose nanocrystals and cellulose nanorods are synonyms (See the review by Klemm et al. Angew. Chem. Int. Ed. 50 (2011), 5438-5466).

Reply: In the original version of the manuscript, the cellulose nanoparticles were classified according to their morphology. However, when the mode of preparation nad degree of crystallinity are taken into account, the terms cellulose nanowhiskers, cellulose nanocrystals and cellulose nanorods are really synonyms, which was added to the Introduction, including the reference “Klemm et al. 2011”.

3) The scale bars in Figure 1 A-C are not visible.

Reply: The Figure 2 with its caption was deleted due to copyright problems (see the replies to the Reviewer 2)

4) Lines 569-570 read: “BNC itself, i.e. without any additives, showed a great capacity to stimulate wound healing, i.e. regeneration of the epidermis and dermis.” But lines 720-721 read: “Unlike bacterial nanocelullose, however, plant cellulose can induce wound healing by itself, i.e. without any additives.” These sentences contradict each other regarding the capacity of bacterial nanocellulose itself to induce wound healing.

Reply: It was a mistake – the sentence was corrected to “Like bacterial nanocelullose, plant cellulose can also appear in the form of a hydrogel containing nanofibrils, and it can induce wound healing by itself, i.e. without any additives.

Originally, the authors wanted to say that “unlike bacterial cellulose, plant nanocellulose can have antimicrobial properties by itself, i.e. without any additives.”

5) Some typos to be corrected are:

- Line 21: Gluconacetobacter

- Line 225: SH-SY5Y

Reply: The typos were corrected.